# VHL synthetic lethality screens uncover CBF-β as a negative regulator of STING

James A. C. Bertlin [1], Tekle Pauzaite[1], Qian Liang[2], Niek Wit [1], James C. Williamson [1], Jia Jhing Sia [1], Nicholas J. Matheson [1,3], Brian M. Ortmann[1,4], Thomas J. Mitchell[5], Anneliese O. Speak [1], Qing Zhang [2] & James A. Nathan [1] ✉

Clear cell renal cell carcinoma (ccRCC) represents the most common form of kidney cancer and is typified by biallelic inactivation of the von Hippel-Lindau (*VHL*) tumour suppressor gene. Here, we undertake genome-wide CRISPR/Cas9 screening to reveal synthetic lethal interactors of *VHL*, and uncover that loss of Core Binding Factor β (CBF-β) causes cell death in *VHL*-null ccRCC cell lines and impairs tumour establishment and growth in vivo. This synthetic relationship is independent of the elevated activity of hypoxia inducible factors (HIFs) in *VHL*-null cells, but does involve the RUNX transcription factors that are known binding partners of CBF-β. Mechanistically, CBF-β loss leads to upregulation of type I interferon signalling, and we uncover a direct inhibitory role for CBF-β at the *STING* locus controlling Interferon Stimulated Gene expression. Targeting CBF-β in kidney cancer both selectively induces tumour cell lethality and promotes activation of type I interferon signalling.

Kidney cancer is a major and growing cause of morbidity and mortality worldwide, with ~180,000 patients dying each year from the disease[1,2]. Clear cell renal cell carcinoma (ccRCC) is the most common histological subtype in adults, accounting for ~85% of cases[3,4]. While low-grade tumours can be cured by partial or radical nephrectomy, the systemic medical options to manage metastatic ccRCC remain largely inadequate, and only 13% of patients with metastatic spread at the time of diagnosis survive 5 years[5].

Biallelic inactivation of the *VHL* tumour suppressor gene at chromosome 3p25 is a hallmark event in the development of ccRCC tumours[6,7]. Loss-of-heterozygosity of chromosome 3p occurs in childhood or adolescence, followed decades later by mutation or epigenetic inactivation of the remaining *VHL* allele. The subsequent acquisition of subclonal driver mutations, including of the chromatin regulators *PBRM1*, *SETD2* and *BAP1*, culminates in the establishment of overt ccRCC tumours[8–10]. Inherited *VHL* mutations also underlie the hereditary autosomal dominant von Hippel-Lindau disease, in which patients develop numerous neoplasms early in life[11].

VHL forms the substrate recognition component of the E3 ligase complex responsible for the ubiquitylation of prolyl-hydroxylated hypoxia inducible factor (HIF)-α subunits under conditions of abundant molecular oxygen[12,13]. In hypoxic environments, or upon genetic depletion of VHL, HIF-α accumulates, heterodimerises with HIF-1β/ARNT, and translocates to the nucleus, whereupon it drives a diverse transcriptional program, including genes implicated in angiogenesis and glycolytic metabolism[14]. The stabilisation of the HIF-2α isoform in particular accounts for many of the distinctive features of ccRCCs: vascular endothelial growth factor (VEGF) secretion results in highly angiogenic tumours, while increased fatty acid biosynthesis and impaired mitochondrial fatty acid transport cause cytoplasmic lipid deposition and the characteristic clear cell morphology[15–18].

The term 'synthetic lethality' is used to describe the phenomenon in which the perturbation of two individually non-essential genes is lethal to a cell[19]. This paradigm has been used to target tumours with characteristic mutational drivers, exemplified by the employment of PARP inhibitors in the treatment of BRCA1/2-defective breast and

[1]Cambridge Institute of Therapeutic Immunology & Infectious Disease (CITIID), Jeffrey Cheah Biomedical Centre, Department of Medicine, University of Cambridge, Cambridge, UK. [2]Simmons Comprehensive Cancer Center, Department of Pathology, University of Texas Southwestern Medical Center, Dallas, TX, USA. [3]NHS Blood and Transplant (NHSBT), Cambridge, UK. [4]Centre for Cancer, Newcastle Centre for Cancer, Paul O'Gorman Building, Newcastle upon Tyne, UK. [5]Early Cancer Institute and Department of Surgery, University of Cambridge, Cambridge, UK. ✉e-mail: jan33@cam.ac.uk

ovarian cancers[20–23]. Over the last two decades, several groups have employed genetic and chemical screening platforms, computational algorithms, and hypothesis-driven approaches to identify synthetic lethal interactors of *VHL*[24–34]. However, while some hits may hold potential for the specific treatment of ccRCC, there has been little concordance between studies, presumably reflecting the diversity of cell lines used and the limitations of existing approaches, including the off-target effects of RNA interference techniques and the use of sub-optimal libraries.

Utilising pooled genome-wide CRISPR/Cas9 mutagenesis in ccRCC cell lines, we uncover synthetic lethal interactions for *VHL*, and find that Core Binding Factor β (CBF-β) loss selectively results in decreased tumour growth of ccRCCs. Mechanistically, CBF-β deficiency promotes cell death in ccRCCs, but we also find an unanticipated role for CBF-β in the repression of type I interferon (IFN) signalling. Loss of CBF-β results in Interferon Stimulated Gene (ISG) expression, resulting from the de-repression of STING (Stimulator of Interferon Genes) transcription. CBF-β therefore tunes cell-intrinsic type I interferon signalling, and its loss sensitises cells to a heightened STING-mediated immune response. Therefore, we propose that CBF-β offers a potential therapeutic target in ccRCC through its synthetic lethal relationship with *VHL* loss.

## Results

### Genome-wide CRISPR/Cas9 screening reveals synthetic viability interactors of *VHL*

To find genes that alter the viability of *VHL*-null cells, we employed the widely-used ccRCC models 786O and RCC4 alongside paired VHL-reconstituted lines, which preferentially express different isoforms of VHL due to the use of alternative translation start sites (786O + *VHL* and RCC4 + *VHL*) (Fig. 1a and Supplementary Fig. 1a)[12,35,36]. Pooled CRISPR/Cas9 screening was performed by transducing isogenic cell lines expressing Cas9 with the Toronto KnockOut version 3.0 (TKOv3) genome-wide sgRNA library, passaging cells until 17 population doublings had occurred, and undertaking deep sequencing of sgRNAs (Fig. 1b and Supplementary Fig. 1b)[37]. Identification of a curated set of core essential genes that dropped out over the duration of the culture confirmed the efficiency of the screen (Supplementary Fig. 1c and Supplementary Data 1)[37]. Synthetic viability interactors of *VHL* were identified as those targeted by sgRNA sequences which were relatively less or more abundant in the *VHL*-null lines at the conclusion of the screen (Fig. 1c and Supplementary Data 1). We used three different analyses to identify genes with a fitness advantage or disadvantage: BAGEL2, DrugZ and MAGeCK[38–40]. The positive controls *ARNT*/*HIF1β* and *MTOR* dropped out in the *VHL*-null setting, confirming the validity of our approach[34,41]. Alongside these expected findings we identified a putative synthetic lethal relationship between *VHL* and *CBFB* (encoding CBF-β) in both ccRCC backgrounds (Fig. 1c, d). CBF-β is best characterised as forming a transcriptional complex with RUNX proteins to activate or repress target genes[42,43]. Both *RUNX1* and *RUNX2* were identified in the RCC4 screen (Fig. 1c), consistent with a potential role of these transcriptional complexes in the synthetic viability relationship with *VHL*.

*KEAP1* was identified as the top hit that when mutagenised conferred a proliferative advantage to *VHL*-null cells (Fig. 1c, d). KEAP1 ubiquitylates and traps NRF2 in the cytoplasm to prevent NRF2-mediated transcription of antioxidant factors including *NQO1* and *HMOX1*[44,45], raising the possibility that upregulation of the NRF2 response may confer a survival advantage following *VHL* loss.

We developed a competitive growth assay with a fluorescent read-out to validate target genes (Fig. 1e). By labelling the 786O *VHL*-null cells with mCherry and the *VHL*-reconstituted cells with GFP, we could readily distinguish a growth advantage or disadvantage when both cell lines were mixed in a 1:1 ratio (Fig. 1e). Inducible depletion of CBF-β,

using both sgRNA and shRNA, confirmed the *VHL*/*CBFB* synthetic lethal relationship (Fig. 1f and Supplementary Fig. 1d, e). Conversely, KEAP1 loss increased the growth of 786O cells relative to 786O + *VHL* cells (Fig. 1g and Supplementary Fig. 1f). We also found that NRF2 was synthetic lethal with VHL in the 786O competitive growth assay (Fig. 1g), but its modulation did not alter the levels of HIF-2α or HIF target gene expression (Supplementary Fig. 1f, g). Therefore, the KEAP1-NRF2 axis may protect against a greater degree of oxidative stress in VHL-deficient cells.

### Deletion of *CBFB* causes cell death in *VHL*-null cells

We chose to focus on *CBFB* given the potential synthetic lethal relationship between *CBFB* and *VHL*, the additional identification of *RUNX* genes in the RCC4 screen, and the observation that CBF-β is abundant in ccRCC tumours (Fig. 2a), with its expression correlated strongly with poor survival outcomes in both The Cancer Genome Atlas (TCGA) and the Clinical Proteomic Tumor Analysis Consortium (CPTAC) databases (Fig. 2b, c).

We first sought to characterise the proliferation phenotype of CBF-β loss in ccRCC cells. The growth and colony formation potential of 786O cells was significantly impaired by CBF-β depletion, but this growth defect was abrogated in the isogenic *VHL*-reconstituted line (Fig. 2d and Supplementary Fig. 2a). CBF-β loss also impeded growth in some other *VHL*-null ccRCC lines (A498 and RCC10 cells), but not in 769P ccRCC cells, or the HKC8 model of healthy proximal tubule epithelium (Supplementary Fig. 2b, c). Prior studies report differing sensitivities to HIF-2α inhibitors in ccRCC lines, and we considered whether an interaction with CBF-β and HIF-2α could explain the varied sensitivity of ccRCC lines to impaired growth with *CBFB* loss[41]. However, the functional interaction between *CBFB* and *VHL* was independent of HIF signalling, as the same phenotype remained in 786O cells following clonal knockout of HIF1β (Supplementary Fig. 2b, c, final columns), and *CBFB* knockout did not consistently affect the levels or transcriptional activity of HIF-2α (Supplementary Fig. 2d, e).

The proliferation defect in 786O cells could be caused by either a defect in cell cycling, or an increased rate of cell death. We excluded a significant deficiency in cell cycling as neither the distribution of cells within cycle phases nor the progression of synchronised populations through the cell cycle were affected by *CBFB* knockout (Fig. 2e and Supplementary Fig. 3a). Rather, the depletion of CBF-β caused cytotoxicity in *VHL*-null cells, demonstrated by SYTOX AADvanced staining and quantification of released lactate dehydrogenase (LDH) (Fig. 2f, g). We investigated the mode of cell death using assays of apoptosis, or specific inhibitors of necroptosis (Nec-1s), pyroptosis (VRT-043198), and ferroptosis (ferrostatin). Apoptosis assays did not demonstrate a clear induction of this pathway following CBF-β depletion (Supplementary Fig. 3b, c), and treatment with the highest tolerable doses of the various cell death inhibitors did not restore growth in the CBF-β-deficient 786O cells (Supplementary Fig. 3d). Finally, we did not observe any effect of *CBFB* knockout on the sensitivity of cells to the vacuolar H⁺-ATPase inhibitor Bafilomycin A1, suggesting that there was no involvement of autophagy-related cell death (Supplementary Fig. 3e). Therefore, while the precise modality of cell death is unclear, our studies indicate that CBF-β loss does lead to cytotoxicity in 786O cells.

To establish whether CBF-β loss also impaired tumourigenesis in vivo, we performed xenograft experiments in immunodeficient NSG mice. First, we observed that CBF-β knockout severely restricted the ability of 786O cells to develop tumours in the subcutaneous microenvironment as tumours were almost undetectable 21 days after injection (Fig. 2h). Next, to interrogate the more clinically-relevant scenario of established kidney tumours, we employed an orthotopic model of ccRCC using luciferase-expressing 786O cells harbouring doxycycline-inducible sgRNA vectors[46]. Upon confirmation of tumour

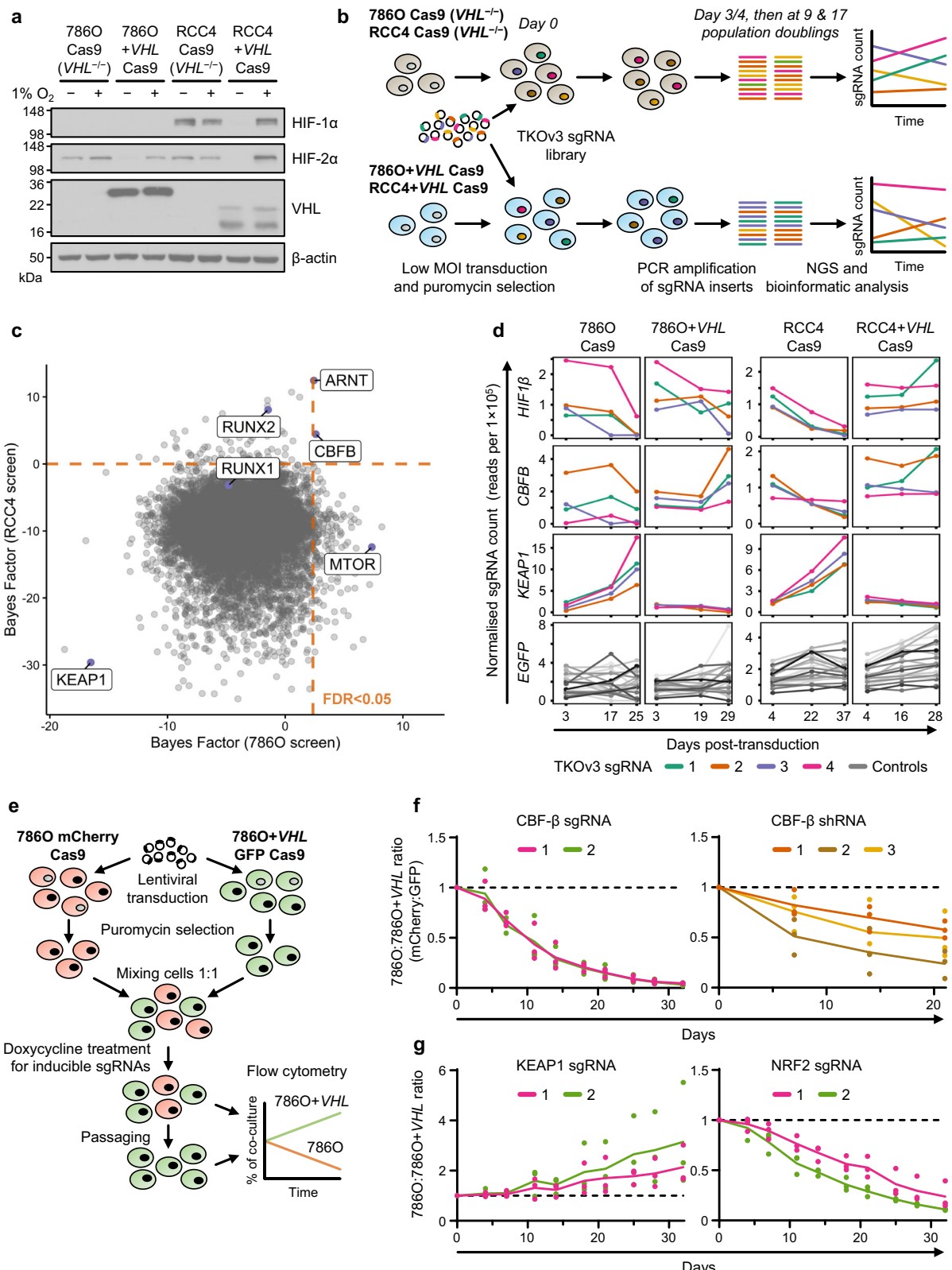

growth in each kidney, as indicated by increased luciferase activity over time, we administered doxycycline chow to mice in order to induce CBF-β depletion in vivo. Deletion of CBF-β with two independent sgRNAs both substantially impeded tumour growth (Fig. 2i–k and Supplementary Fig. 3f) and prevented metastasis to the lungs (Fig. 2l, m), confirming the relevance of the CBF-β/VHL synthetic lethality in vivo.

## CBF-β and RUNX proteins are required for the synthetic lethal interaction with VHL

Given the canonical transcriptional function of CBF-β in complex with RUNX1-3, and the manifestation of RUNX2 as a synthetic lethal hit in the RCC4 CRISPR screen (Fig. 1c), we hypothesised that *CBFB* loss could result in the death of *VHL*-null cells via impaired RUNX activity. Indeed, the combined knockout of RUNX1 and RUNX2, but not of

**Fig. 1 | Genome-wide CRISPR/Cas9 screening reveals synthetic viability interactors of *VHL*. a** HIF-α levels in ccRCC cells and paired *VHL*-reconstituted cells after culture at either 1% or 21% $O_2$ for 24 h. Immunoblot representative of 3 independent experiments. **b** Schematic of CRISPR/Cas9 screen design. Paired *VHL*-null and -reconstituted cell lines were transduced with the genome-wide TKOv3 sgRNA library. Cells were passaged in parallel for several weeks, and genomic DNA extracted at early, intermediate and late timepoints for the identification of sgRNA abundance by Next Generation Sequencing (NGS). sgRNAs targeting synthetic lethal interactors of *VHL* are selectively depleted in the *VHL*-null 786O Cas9 and RCC4 Cas9 backgrounds. **c** Pairwise analysis of the late timepoints of 786O Cas9 vs. 786O + *VHL* Cas9 cells (x-axis) and RCC4 Cas9 vs. RCC4 + *VHL* Cas9 cells (y-axis) from CRISPR/Cas9 screens. A higher Bayes Factor calculated by BAGEL2 indicates a more robust synthetic lethal relationship between the given gene and *VHL*. Orange lines denote FDR < 0.05. **d** Normalised sgRNA counts from CRISPR/Cas9 screens.

Guides targeting *EGFP* serve as negative controls. **e** Competitive growth assay method. Transduced fluorophore-expressing 786O and 786O + *VHL* cells are mixed 1:1, passaged and harvested for flow cytometry analysis. 100 ng/ml doxycycline is added to experiments involving inducible sgRNAs to initiate gene editing. **f, g** Validation of functional interactions between *VHL* and *CBFB* (**f**), or between *VHL* and *KEAP1* and *NRF2* (**g**), by competitive growth assay using doxycycline-inducible sgRNAs or constitutively-expressed shRNAs. *n* = 3 biologically independent replicates. Curves connect mean value for each timepoint. One-way ANOVA based on Area Under the Curve compared to empty vector or scrambled control: CBF-β sgRNA 1 *P* < 0.0001; CBF-β sgRNA 2 *P* < 0.0001; CBF-β shRNA 1 *P* = 0.14; CBF-β shRNA 2 *p* = 0.0034; CBF-β shRNA 3 *P* = 0.042; KEAP1 sgRNA 1 *P* = 0.42; KEAP1 sgRNA 2 *P* = 0.11; NRF2 sgRNA 1 *P* = 0.0002; NRF2 sgRNA 2 *P* < 0.0001. Source data are provided as a Source Data file.

either isoform individually, caused a similar degree of synthetic lethality to CBF-β loss in a competitive growth assay (Fig. 3a), and the deletion of all three RUNX proteins mimicked the observed CBF-β proliferation phenotype in 786O cells (Fig. 3b). However, the synthetic lethal effect of CBF-β with VHL loss could not be simply explained by a reduction of complex formation with RUNX proteins, as we noted that deletion of CBF-β caused a profound decline in RUNX1 and RUNX2 protein abundance in both 786O and 786O + *VHL* cells (Fig. 3c, d).

To further interrogate the role of CBF-β/RUNX dimerisation, we expressed FLAG-tagged overexpression constructs of wild-type (WT) CBF-β and an N104A-mutant form, which fails to bind the Runt homology domain of RUNX proteins[47], in 786O or CBF-β-deficient 786O cells. We confirmed that the N104A mutation prevented RUNX binding by co-immunoprecipitation (Fig. 3e). Importantly, the sgRNA used to induce CBF-β deletion (CBF-β sg2) was directed against the boundary between intron 2 and exon 3 of the *CBFB* locus and therefore permitted knockout of the endogenous CBF-β protein while sparing the overexpression constructs. However, overexpression of either wild-type or N104A CBF-β restored the proliferation of CBF-β-null 786O cells to a similar extent (Fig. 3f, g). Moreover, we noted both CBF-β constructs restore RUNX protein levels without increasing *RUNX1-3* mRNA expression (Fig. 3h, i). Together, these findings indicate that while RUNX proteins are required for the *CBFB* synthetic lethal growth defect in 786O cells, the direct association of CBF-β with RUNX proteins may not be necessary.

### *CBFB* loss induces a cell-intrinsic interferon response

As CBF-β and RUNX appeared to exercise their functions at both transcriptional and translational levels, we undertook RNA sequencing (RNA-Seq) and tandem mass tag (TMT)-labelled mass spectrometry to characterise the effects of CBF-β loss in VHL-proficient and -deficient backgrounds (Fig. 4a). Strikingly, in 786O cells, *CBFB* knockout caused a net increase in transcription and specifically induced the expression of a plethora of ISGs exclusively in *VHL*-null 786O cells (Fig. 4b–d and Supplementary Data 2, 3). Quantitative PCR (qPCR) analysis confirmed that the upregulation of several ISGs (*IFIT1*, *OASL*, *ISG15*, and *RSAD2*) was specific to CBF-β depletion, using both sgRNA and shRNA (Supplementary Fig. 4a–c), and prevented by reconstitution with both wild-type and N104A-mutant CBF-β-FLAG constructs (Fig. 4e and Supplementary Fig. 4d). The selective ISG stimulation in *VHL*-null cells was also independent of HIF activity, as it was not abolished by *HIF1β* knockout (Fig. 4f), and observed in other but not all ccRCC lines (Supplementary Fig. 4e).

We next determined how the induction of ISGs in CBF-β-deficient 786O and 786O + *VHL* cells compared with type I IFN treatment by titrating levels of IFN-β. We observed that CBF-β loss was approximately equivalent to treatment with 24 h of 10 IU/ml IFN-β (Fig. 4g). CBF-β loss also induced ISGs non-uniformly, affecting a subset of IFN-β-responsive genes (IFI44, ISG15, and IRF9), but not MDA5 or MX1 (Fig. 4g). Given this low amplitude ISG response, we considered

whether the induction of ISGs related to lentiviral transduction of our sgRNA targeting *CBFB*. However, ISGs were not induced if cells were transduced with sgRNAs of similar G:C content targeting alternative loci to *CBFB* (Supplementary Fig. 4f). Thus, CBF-β loss promotes an ISG response in 786O cells.

To further explore ISG expression in VHL-deficient renal cancer, we analysed the single-cell transcriptional profiles of treatment-naïve renal tumours from 12 patients who underwent surgical resection[48]. Using the same type I IFN gene signatures as for the bulk RNA-Seq (Fig. 4b, c), we observed that there was considerable heterogeneity in ISG expression across cell populations within tumours (Fig. 4h and Supplementary Fig. 5). RCC cells showed only low levels of ISGs, with the bulk of ISG expression stemming from the endothelial cells and certain immune subsets (Fig. 4h and Supplementary Fig. 5). *CBFB* transcript levels were low across all cell types, therefore correlating *CBFB* expression with ISG expression was not possible. However, these transcriptional profiles are consistent with general repression of ISGs within the tumour cells.

### A STING-TBK1-IRF3 axis drives ISG expression in VHL-deficient renal cancer cells

The finding that *CBFB* loss resulted in type I IFN expression suggested either that CFB-β negatively regulates IFN signalling or that its loss results in damage signals detected by pattern recognition receptors (PRRs). ISG expression is classically stimulated when microbial nucleic acids are sensed by PRRs which converge, via the adaptors STING, MAVS and TRIF, on the TBK1/IKK-ε kinases to promote the phosphorylation and dimerisation of IRF3[49–51], activating ISGs and type I IFN expression (Fig. 5a). Upon secretion, type I IFNs then act in a paracrine or autocrine manner to initiate a JAK-STAT phosphorylation cascade which culminates in the transcription of ISGs by STAT1/2 heterodimers (in association with IRF9) or, less commonly, STAT1 homodimers (Fig. 5a)[52,53]. Our finding that *CBFB* deletion only altered the growth of 786O but not 786O + *VHL* cells in co-culture (Fig. 1f) suggested that type I IFN secretion was unlikely to explain the induction of ISGs, and that a cell-intrinsic pathway of direct ISG stimulation was involved. We therefore anticipated that elimination of STAT1/2 signalling would abolish the IFN response to CBF-β loss. To our surprise, however, knockout of *STAT1*, *STAT2* or *IRF9* individually, or in combination, did not prevent *IFIT1* induction (Fig. 5b and Supplementary Fig. 6a–c), and we also did not observe phosphorylation of STAT1 at Tyr701 with *CBFB* knockout (Supplementary Fig. 6d).

We therefore examined if CBF-β loss acted upstream in ISG induction by depleting IRF3, as IRF3 homodimers have been reported to directly bind the IFN-stimulated response elements (ISREs) of canonical ISGs to stimulate their transcription independently of interferons[54–56]. *IFIT1* expression was profoundly sensitive to *IRF3* knockout in the CBF-β-null cells (Fig. 5c and Supplementary Fig. 6e). Moreover, the ISG response observed upon CBF-β knockout was entirely dependent on IRF3, and not on the closely-related, immune

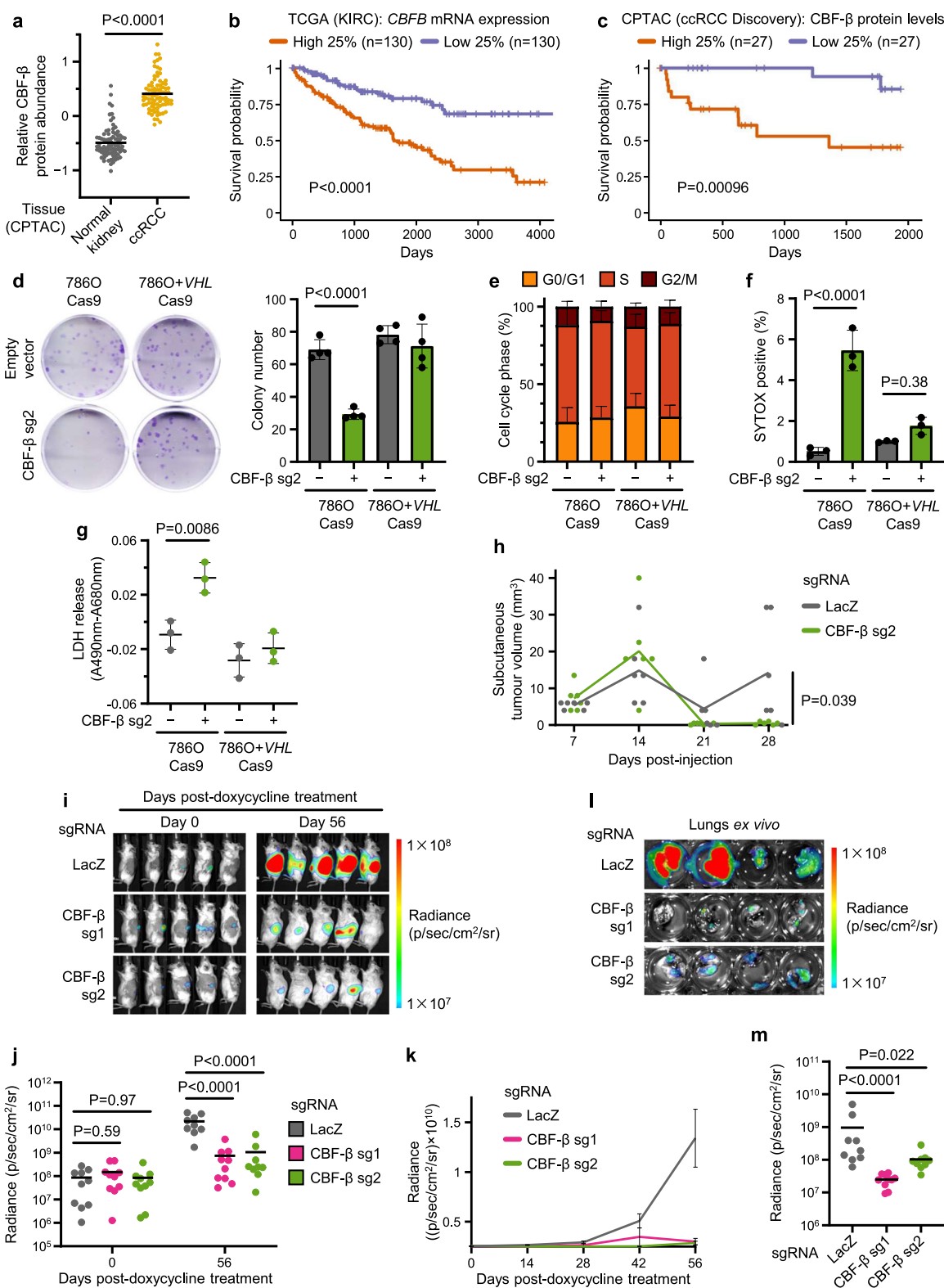

cell-specific IRF7 (Fig. 5c and Supplementary Fig. 6e)[57], and was associated with phosphorylation of IRF3 at Ser386 (Fig. 5d and Supplementary Fig. 6f), which is essential for IRF3 activation[58]. To confirm that IRF3 phosphorylation was required for subsequent ISG expression, we treated CBF-β-null 786O cells with the TBK1 inhibitor, GSK8612. This treatment prevented IRF3 Ser386 phosphorylation and the subsequent ISG induction in a dose-dependent manner (Fig. 5d, e). Chromatin

immunoprecipitation (ChIP) experiments showed some increased binding of IRF3 to ISREs upon CBF-β depletion (Supplementary Fig. 6g). Lastly, we established that STING, but not MAVS or TRIF, was required for the induction of ISGs following *CBFB* knockout (Fig. 5f, g and Supplementary Fig. 6h), identifying a STING-TBK1-IRF3 axis that is activated in a CBF-β loss-dependent manner.

**Fig. 2 | CBF-β loss causes cell death in VHL-null ccRCCs and impairs tumour growth in vivo. a** Relative CBF-β protein expression in ccRCC samples and matched normal kidney tissue within the CPTAC ccRCC Discovery dataset[126,127]. *n* = 84 individuals. Mean denoted by horizontal bar. Paired two-sided *t*-test. **b, c** Kaplan–Meier survival analysis of TCGA (**b**) and CPTAC (**c**) ccRCC data[7,126,127], comparing tumours in the highest and lowest quartiles of *CBFB* mRNA (**b**) or CBF-β protein (**c**) expression. *n* = 130 (**b**) or 27 (**c**) individuals per group. Log-rank test. **d** Cells transduced with CBF-β sg2 or an empty vector control analysed by clonogenic assay. Representative image of crystal violet-stained colonies. *n* = 4 biologically independent replicates. Mean ± SD. Two-way ANOVA. **e** Proportion of cells in the indicated cell cycle phases in asynchronous cell populations transduced with CBF-β sg2 or a control, determined by BrdU incorporation and propidium iodide staining. *n* = 4 biologically independent replicates. Mean ± SD. **f** SYTOX AADvanced cell death assay of 786O Cas9 and 786O + *VHL* Cas9 transduced with CBF-β sg2 or a control. *n* = 3 biologically independent replicates. Mean ± SD. Two-way ANOVA. **g** LDH in culture supernatants in cells transduced with CBF-β sg2 or a control. *n* = 3 biologically independent replicates. Mean ± SD. Two-way ANOVA. **h** Tumour volume following subcutaneous injection of 786O Cas9 cells transduced with sgRNAs targeting CBF-β or LacZ into NSG mice. Tumours in this model show an initial growth before regressing and finally adopting a sustained growth trajectory after 21 days[124]. *n* = 6 mice per group. Curves connect mean value for each time-point. Two-sided Mann–Whitney U test of tumour volumes at day 28 post-injection. **i–m** Xenograft model of luciferase-expressing 786O Cas9 cells injected orthotopically into the left kidneys of NSG mice, using doxycycline treatment to induce expression of sgRNAs targeting CBF-β (sg1 and sg2) or LacZ as a control. Representative bioluminescence imaging of mice at day 0 and day 56 after the initiation of doxycycline treatment (**i**). Quantification of the bioluminescence of primary tumours following doxycycline treatment (**j, k**). Representative and quantified bioluminescence of isolated lungs harvested from mice at day 56 post-doxycycline treatment (**l, m**). *n* = 10 mice per group, of which 1 died between days 42 and 56 in the LacZ and CBF-β sg2 groups. Mean ± SEM. One-way ANOVA. Source data are provided as a Source Data file.

As type I IFN activation can induce a plethora of cellular phenotypes, including senescence, apoptosis and cell cycle arrest, we next considered whether the activation of the STING-TBK1-IRF3 axis was sufficient to explain the synthetic lethal phenotype observed between *CBFB* and *VHL*, but STING or IRF3 depletion by themselves did not fully reverse the synthetic interaction (Supplementary Fig. 6i–k). However, IFN-β treatment did mildly increase cell death in *VHL*-null 786O cells compared to immortalised renal tubular epithelial cells and *VHL*-reconstituted 786O cells (Fig. 5h and Supplementary Fig. 6l). Therefore, while activation of the IRF3 axis cannot fully explain the synthetic lethal phenotype, ccRCCs are sensitive to type I IFN-induced cell death.

### CBF-β represses STING to tune the type I interferon response

We were intrigued by the notion that CBF-β may be a negative regulator of STING. We noted that not only did CBF-β loss result in the STING-dependent induction of ISGs, but that protein levels of STING increased in CBF-β-null 786O cells (Fig. 5g). Therefore, we tested whether CBF-β depletion altered STING expression and the sensitivity of the cGAS-STING pathway to double stranded DNA (dsDNA) (Fig. 6a). Firstly, in 768O cells, we observed that *STING* transcription was dramatically increased upon *CBFB* knockout but reduced by its overexpression (Fig. 6b). Moreover, using herring testes DNA (HT-DNA), a long 'non-self' DNA molecule which enables specific interrogation of cGAS-STING function (Fig. 6c)[59,60], we found a striking synergistic induction of ISGs following CBF-β loss: while HT-DNA alone provided only modest *IFIT1* upregulation, we observed a dramatic induction in response to combined HT-DNA transfection and CBF-β sg2 transduction (Fig. 6d). This was accompanied by increased expression of *IFNB1* (encoding IFN-β), despite its transcription not being affected by *CBFB* knockout alone. Conversely, overexpression of the wild-type CBF-β-FLAG construct blunted the response to HT-DNA (Fig. 6d).

It was plausible that CBF-β loss may contribute to STING activation through the release of dsDNA into the cytoplasm from either a mitochondrial or genomic source[61–67]. However, we found no evidence of mitochondrial dysfunction or superoxide generation by flow cytometry in 786O cells (Supplementary Fig. 7a, b), and neither CBF-β deletion nor overexpression affected the cellular oxygen consumption rate (Supplementary Fig. 7c). There was no increase in the leakage of DNA from mitochondria upon *CBFB* knockout, although interestingly this was reduced in VHL-reconstituted 786O cells (Supplementary Fig. 7d). This is consistent with a recent report that *VHL* loss increases mitochondrial dsDNA leakage, and may partly explain why *VHL*-null cells are particularly responsive to CBF-β-mediated STING induction[68]. Moreover, there was no overt genomic DNA damage or expression of ccRCC-associated endogenous retroviruses (ERVs) in response to transduction with CBF-β sg2 (Supplementary Fig. 7e–h and Supplementary Data 4)[69–72]. Indeed, while a basal level of cGAMP was required for STING activity, cGAS knockout caused only a minor reduction in CBF-β knockout-induced STING expression, confirming that the modulation of STING levels by CBF-β is largely independent of interferon signalling (Fig. 6e and Supplementary Fig. 7i). We noted that while overexpression of STING alone was insufficient to induce ISG expression, it did sensitise cells to stimulation with dsDNA (Supplementary Fig. 8a). Overall, therefore, our results indicated that CBF-β-mediated regulation of STING levels tunes the cell-intrinsic ISG response, but that an additional stimulus is required to provoke ISG expression in the absence of CBF-β, the precise identity of which remains unclear. In support of this, equivalent experiments confirmed the synergistic induction of ISGs and *IFNB1* upon CBF-β depletion and HT-DNA transfection in several other cell lines, including those of non-cancerous renal origin (Supplementary Fig. 8b–d).

The involvement of CBF-β in tuning *STING* transcription implied that it may be tonically repressed by CBF-β/RUNX dimers. To interrogate this, we performed ChIP sequencing (ChIP-Seq) of CBF-β in 786O cells using wildtype and overexpressed CBF-β (Fig. 7a). CBF-β was found to interact robustly at the *STING* locus, associated with a 5'-YGYGGTY-3' RUNX consensus motif at the 3'-end of the gene[73,74]. Furthermore, analysis of published CBF-β chromatin binding patterns in SAOS2 cells revealed a strong peak at an alternative RUNX consensus motif near the transcription start site of the gene (Fig. 7b)[75]. Notably, we did not detect any significant binding at *IFIT1* or other ISGs (Fig. 7a). ChIP-qPCR experiments validated the binding of CBF-β at both ends of the *STING* gene in 786O cells (Fig. 7c, d).

Next, we sought to identify the extent to which RUNX proteins are involved in this process of STING regulation. As for CBF-β itself, the overexpression of RUNX1, and to a lesser extent RUNX2, represses STING transcription (Fig. 7e, f). In addition, RUNX1 overexpression can overcome the effect of CBF-β loss on downstream ISG expression (Fig. 7e, f). Together, these findings indicate that CBF-β and RUNX transcription factors tune the level of STING through transcriptional repression. RUNX1 appears to play a dominant role in this process but requires CBF-β to exert a maximal repressive effect.

Finally, we explored whether the tonic regulation of STING by CBF-β has broader implications for type I interferon responses. CBF-β is implicated in human/simian immunodeficiency virus (HIV/SIV) infection, whereby the virallyencoded Vif associates with CBF-β to hijack a ubiquitin E3 ligase complex and degrade host APOBEC3G[76,77]. Besides disabling this important layer of antiviral defence, the trapping of CBF-β by Vif in the cytoplasm also prevents it from co-operating with RUNX transcription factors and exerting transcriptional activity (Fig. 7g)[78]. We therefore expressed GFP-Vif in 786O cells, and measured basal and dsDNA-induced type I interferon signalling. Vif expression increased ISG expression, similarly to CBF-β loss (Fig. 7h), and triggered the expression of both STING and downstream ISGs (Fig. 7h, i). Together, our findings establish CBF-β as a core and direct regulator of STING-mediated interferon signalling, with broad physiological and

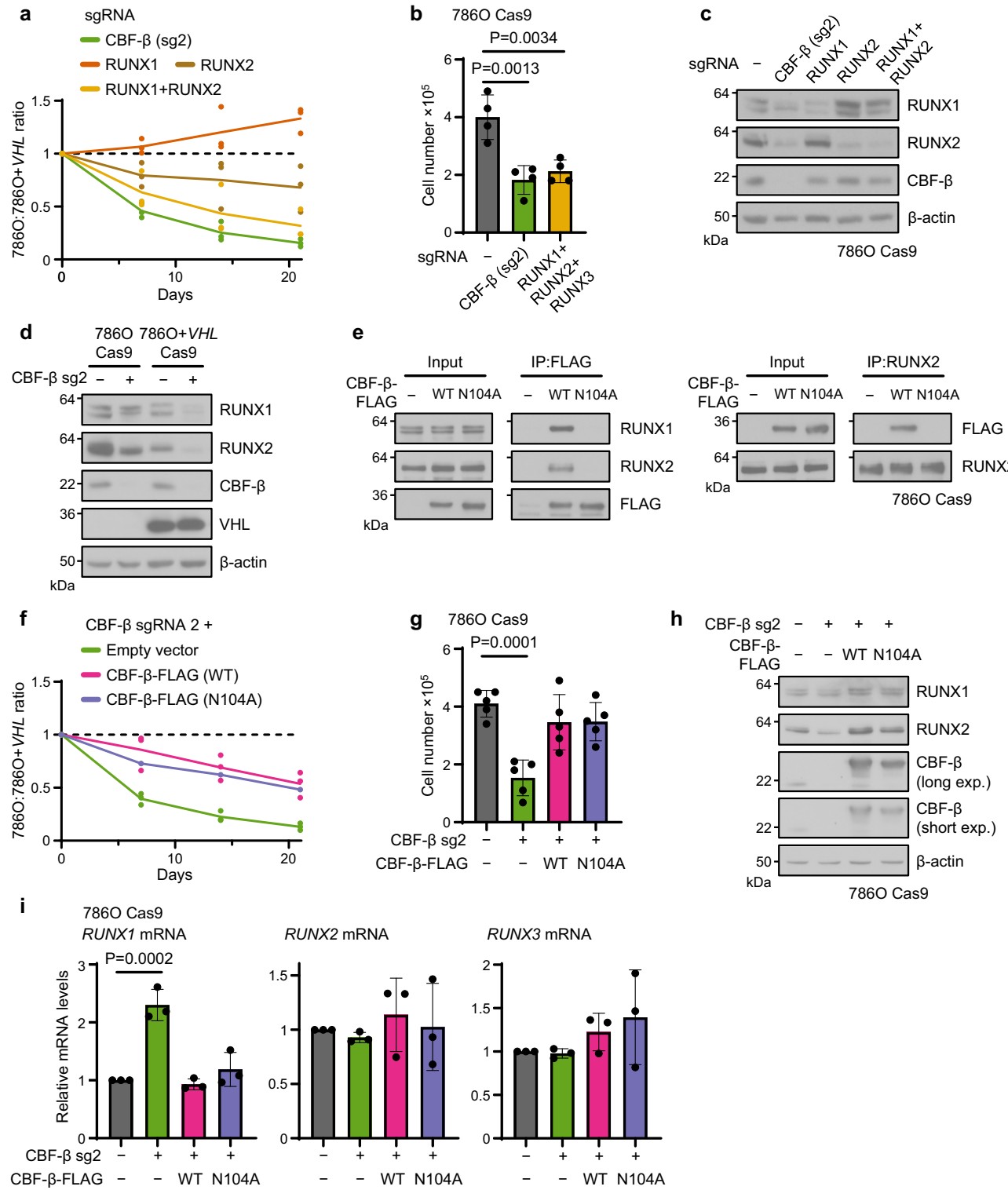

pathological implications in the context of anti-retroviral immunity and kidney cancer (Fig. 8).

## Discussion

The synthetic lethal interaction between *VHL* and *CBFB* provides a new route to target ccRCCs, and is supported by our murine xenograft models in which tumour growth is markedly inhibited. The lethality phenotype following *CBFB* loss is consistent both with observations made upon depletion of RUNX1 and RUNX2 in human kidney cancer lines and in a genetically-engineered mouse (GEM) model[79], and with

CBF-β sitting among the top hits of a computational screening platform to identify synthetic lethal interactions in cancers[24]. Prior studies have also observed that high RUNX2 expression in kidney cancer is associated with a mesenchymal phenotype which promotes tumour cell migration, invasion and proliferation in vitro, and with higher tumour grade in vivo[80–82]. Similarly, elevated RUNX1 in tumour biopsies correlates with poor survival, which has been attributed to altered transcription of extracellular matrix components and remodellers[79,83]. Therefore, although there has been relatively poor agreement between prior VHL-associated synthetic lethality datasets to date[24–31,34], there is

**Fig. 3 | RUNX1 and RUNX2 are regulated post-transcriptionally by CBF-β and contribute to the lethality phenotype. a** Combined loss of *RUNX1* and *RUNX2* functionally mimics *CBFB* deletion. Competitive growth assay of cells transduced with sgRNAs targeting CBF-β, RUNX1, RUNX2, or both RUNX1 and RUNX2. $n = 3$ biologically independent replicates. Curves connect mean value for each timepoint. One-way ANOVA based on Area Under the Curve compared to empty vector control: CBF-β sgRNA 2 $P = 0.0003$; RUNX1 sgRNA $P = 0.31$; RUNX2 sgRNA $P = 0.12$; RUNX1 sgRNA + RUNX2 sgRNA $P = 0.0024$. **b** Proliferation of 786O Cas9 cells transduced with sgRNAs targeting CBF-β or RUNX1, RUNX2 and RUNX3 in combination, or a control vector. $n = 4$ biologically independent replicates. Mean ± SD. One-way ANOVA. **c** Immunoblot of cells transduced with sgRNAs targeting CBF-β, RUNX1, RUNX2, or both RUNX1 and RUNX2. Representative of 3 biologically independent replicates. **d** Immunoblot of 786O Cas9 and 786O + *VHL* Cas9 cells transduced with CBF-β sg2 or a control vector. Representative of 3 biologically independent replicates. **e** Immunoprecipitation (IP) of FLAG-tagged CBF-β and endogenous RUNX2 from cells transduced with vectors encoding CBF-β-FLAG (WT), CBF-β-FLAG (N104A), or an empty vector. Immunoblots representative of 3 biologically independent replicates. **f** Competitive growth assay of cells transduced with CBF-β sg2 and overexpression vectors encoding CBF-β-FLAG (WT), CBF-β-FLAG (N104A), or an empty vector control. $n = 3$ biologically independent replicates. Curves connect mean value for each timepoint. One-way ANOVA based on Area Under the Curve compared to empty sgRNA vector control: CBF-β sgRNA 2 + empty vector: $P = 0.0002$; CBF-β sgRNA 2 + CBF-β-FLAG (WT) $P = 0.066$; CBF-β sgRNA 2 + CBF-β-FLAG (N104A) $P = 0.017$. One-way ANOVA based on Area Under the Curve compared to CBF-β sgRNA 2: CBF-β sgRNA 2 + CBF-β-FLAG (WT) $P = 0.0048$; CBF-β sgRNA 2 + CBF-β-FLAG (N104A) $P = 0.017$. **g** Proliferation of 786O Cas9 cells transduced with CBF-β sg2 and overexpression vectors encoding CBF-β-FLAG (WT) or CBF-β-FLAG (N104A), or an empty vector control. $n = 4$ biologically independent replicates. Mean ± SD. One-way ANOVA. **h, i** 786O Cas9 cells were transduced with CBF-β sg2 and overexpression vectors encoding CBF-β-FLAG (WT) or CBF-β-FLAG (N104A), or an empty vector control. RUNX protein and mRNA levels were analysed by immunoblot (**h**) and qPCR (**i**). $n = 3$ biologically independent replicates. Mean ± SD. One-way ANOVA. Source data are provided as a Source Data file.

now compelling evidence that CBF-β, along with RUNX proteins, is an emerging therapeutic target for ccRCC.

Whilst both CBF-β and RUNX proteins exert synthetic lethal effects with *VHL* loss, the formation of a heterodimeric complex may not be required, as CBF-β overexpression increased RUNX1/2 levels irrespective of direct binding (Fig. 3). This presumed translational effect is consistent with earlier work on CBF-β and RUNX levels in breast cancer cells[84]. The ability of CBF-β to control RUNX levels may also explain prior observations in *Cbfb*-null mouse embryos, in which Runx1 protein was barely detectable despite these embryos displaying similar levels of Runx1 mRNA to *Cbfb*-proficient and -heterozygous controls[85]. Our data suggest, therefore, that small molecule inhibitors of the CBF-β/RUNX interaction would be insufficient to inhibit ccRCC growth[86], and that direct targeting of RUNX-mediated transcription may be more fruitful. To this end, Kamikubo and colleagues have recently developed pyrrole-imidazole polyamides which selectively target the RUNX consensus motif[87]. These have shown antitumour effects in a number of pre-clinical models[88–91], although their safety and specificity remains unclear and particular concerns surround the risk of myelodysplasia given the established role of RUNX1 as a tumour suppressor in haematological lineages[92].

There is also compelling evidence from other systems that RUNX can exert CBF-β-independent roles[93–96], and that post-translational modifications of RUNX isoforms can generate context-specific switches between transcriptional activation and repression[43,97–101]. Interestingly, we detected a net increase in global transcription in our RNA sequencing studies upon CBF-β deletion, which was abrogated by *VHL* re-expression. If the primary function of CBF-β in 786O cells is indeed the regulation of RUNX protein abundance, this may indicate that *VHL* loss causes RUNX to adopt a more repressive state. Further dissection of such pathways, which modulate RUNX function, could therefore unveil new therapeutic targets to specifically modulate cell survival and STING expression in ccRCC while sparing the critical role of RUNX factors in other lineages.

Kidney cancer cell proliferation relies on the suppression of tumour cell-intrinsic type I IFN activity, and our findings that disruption of CBF-β function can activate a cell-intrinsic ISG response in ccRCC cells provide avenues to overcome this repression. Large deletions within chromosomes 9p and 14q are associated with metastasis and poor prognosis in ccRCC[102], and the loss of a cluster of type I IFN genes within the affected chromosomal region 9p21.3 has separately been linked to impaired immune surveillance and increased metastasis[103,104]. In addition, mutations in the tumour suppressors *BAP1*, *PBRM1* and *SETD2*, which co-localise with *VHL* on chromosome 3p and are frequently disrupted in ccRCC, suppress ISGF3-mediated ISG expression[105–107]. A recent study highlighted the importance of the cGAS-STING pathway in ccRCC, proposing that suppression of this pathway is critical to the persistence of highly aneuploid tumour regions by impairing T effector function and myeloid inflammation[108]. Therefore, we postulate that targeting CBF-β could enhance tumour regression in response to immune checkpoint inhibition, as has been observed in trials upregulating type I IFN responses through synthetic cyclic dinucleotides and STING agonists[109,110]. Ongoing work seeks to test this hypothesis by depleting CBF-β in an immunocompetent GEM model.

Since the discovery of IRF3 activation by STING, it has been evident that modulation of STING alone, irrespective of cGAS, can be sufficient in some contexts to stimulate IRF3 phosphorylation[111,112]. Yet despite this, it is striking that only a handful of studies have interrogated the *STING* locus in health and disease to reveal its control by CREB, c-Myc and NF-κB[113–115]. While RUNX1 has been previously shown to attenuate interferon signalling in granulocyte–monocyte progenitors through an unclear mechanism[116], our studies now implicate RUNX/CBF-β as a rheostat of STING levels. This axis could be exploited not only in cancer but also in the context of a wide range of interferonopathies, including those triggered by pathological sensing of cytosolic 'self' DNA or by activating mutations in *STING* itself. Moreover, as STING-driven interferon signalling is an important factor in the clearance of HIV-1[117,118], the regulation of the STING locus by RUNX/CBF-β may have evolved within mammalian cells as a method to counteract its sequestration by Vif. Understanding how RUNX/CBF-β and other factors co-operate to tune the balance between appropriate and pathological STING expression remains an important avenue for further research, with wide-ranging implications for the innate immune control of tumours and for understanding host responses to viral infections.

# Methods

## Cell lines
Cell lines and reagents were sourced as detailed in Supplementary Table 1. 786O, 786O + *VHL* and 769P cells were cultured in Roswell Park Memorial Institute 1640 medium (RPMI-1640), and A498, HEK293T, HKC8, HK2, RCC4, RCC4 + *VHL* and RCC10 cells were cultured in Dulbecco's Modified Eagle Medium (DMEM). All media was supplemented with 10% Fetal Calf Serum (FCS). Cells were maintained at 37 °C in 5% $CO_2$. 2 µg/ml or 2.5 µg/ml Puromycin, 5 µg/ml Blasticidin, or 200 µg/ml Hygromycin was added to the media for antibiotic selection of transduced and transfected cells. Hypoxic culture was undertaken in a Whitley H35 Hypoxystation (Don Whitley) set at 1% $O_2$, 5% $CO_2$ and 94% $N_2$. Cells were confirmed as mycoplasma-free with the MycoAlert detection kit (Lonza Cat#LT07-318), and authenticated by short tandem repeat profiling (Eurofins Genomics).

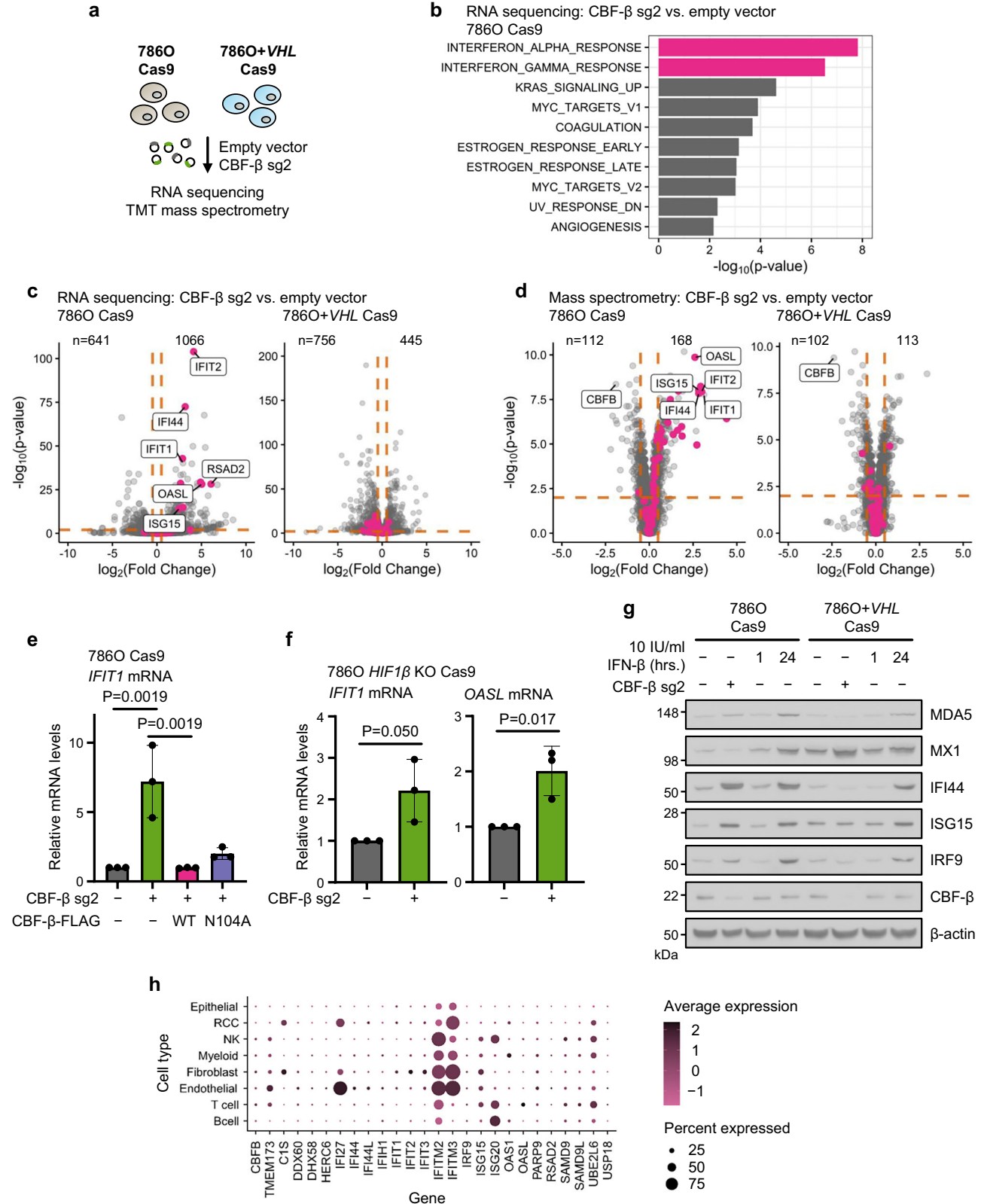

**Nature Communications**|(2026)17:3841

## Plasmids

Cloning of novel gene expression plasmids was performed using Gibson assembly (NEB) of PCR amplicons and restriction digested backbone vectors, using the DNA sequences indicated in Supplementary Table 2. Plasmids were amplified in NEB 5α Competent *E. coli* (Cat#C2987H) grown at 37 °C in LB containing appropriate antibiotics. The sources of plasmids used in this study are indicated in Supplementary Table 1.

## Lentiviral production and transduction

Lentivirus was produced by transfection of HEK293T cells with the relevant vector, pMD.G, and pCMVR8.91 in six-well plates or 15 cm dishes at 60–70% confluence using Fugene 6 Transfection Reagent and Opti-MEM I. Viral supernatant was collected after 48 h through a 0.45 μm filter and stored at −80 °C. For small-scale transductions, 200 μl virus was applied to $5 \times 10^4$ cells in 24-well plates containing

**Fig. 4 | CBF-β loss induces an Interferon Stimulated Gene (ISG) signature.**
**a** 786O Cas9 and 786O + *VHL* Cas9 cells were analysed by RNA sequencing and tandem mass tag (TMT)-labelled LC-MS to assess global changes in mRNA and protein upon CBF-β depletion relative to empty vector-transduced cells. **b** Gene Set Enrichment Analysis of Hallmark gene sets from RNA sequencing analysis of differential gene expression in 786O Cas9 cells following *CBFB* knockout[128]. *n* = 3 biologically independent replicates. Adaptive multi-level split Monte-Carlo method[129]. Volcano plots of RNA sequencing (**c**) and mass spectrometry (**d**) data in the conditions outlined in (**a**). Orange lines indicate log₂(Fold Change) ± 0.5, and $P_{adj}$ = 0.01. Pink dots highlight genes represented within the Hallmark INTER-FERON_ALPHA_RESPONSE set. Number of genes upregulated or downregulated (determined by the log₂(Fold Change) and $P_{adj}$ cut-off values above) are specified. Data points represent the mean value for each gene from 3 biologically independent replicates. Two-sided Wald test (**c**), Benjamini-Hochberg-adjusted moderated

*t*-test (**d**). **e** *IFIT1* is specifically upregulated by CBF-β loss. qPCR analysis of *IFIT1* expression in 786O Cas9 cells upon transduction with CBF-β sg2 and overexpression vectors encoding CBF-β-FLAG (WT) or CBF-β-FLAG (N104A), or an empty vector control. *n* = 3 biologically independent replicates. Mean ± SD. One-way ANOVA. **f** qPCR analysis of ISGs in clonal 786O *HIF1β* knockout Cas9 cells upon CBF-β sg2 transduction, relative to empty vector-transduced controls. *n* = 3 biologically independent replicates. Mean ± SD. Unpaired two-sided *t*-test. **g** Immunoblot of ISGs in 786O Cas9 and 786O + *VHL* Cas9 cells transduced with CBF-β sg2 or an empty vector control. Controls were treated with 10 IU/ml IFN-β for 1 or 24 h. Representative of 3 biologically independent replicates. **h** Analysis of single-cell transcriptomic data from patients with ccRCC[48]. The average expression and the percentage of cells that express type I interferon genes are shown for the principal cell types within the tumour micro-environment. Source data are provided as a Source Data file.

---

500 μl media, and appropriate antibiotic selection performed after 48 h until all untransduced control cells had died.

Lentiviral expression constructs were used for gene overexpression. Most overexpression assays were performed 10–14 days after transduction in comparison to controls transduced with the pHRSIN-SFFV-Puro empty vector. For STING-ALFA expression, cells transduced with pCW57.1-STING-ALFA or an empty vector were treated with 1 μg/ml doxycycline for 24 h before harvesting.

### CRISPR-Cas9 targeted gene deletions
Gene-specific sgRNA sequences listed in Supplementary Table 2 were designed using the E-CRISP or Vienna Biocenter algorithms[119,120]. For cloning into the pKLV-U6gRNA(BbsI)-PGKpuro2ABFP, pKLV-U6gRNA(BbsI)-PGKblast2ABFP and pSpCas9(BB)-T2A-Puro vectors, sgRNA oligonucleotides were designed with appropriate overhangs and cloned according to the Zhang lab protocol[121]. Cloning into tet-pLKO-sgRNA-puro for doxycycline-inducible sgRNA expression was performed using modified overhangs using the method of Huang et al.[122]. Multiplexed expression of sgRNAs targeting RUNX1, RUNX2 and RUNX3, or STAT1 and STAT2 from the same pKLV-U6gRNA(BbsI)-PGKpuro2ABFP vector was achieved by PCR amplification of the tRNA scaffold of pCFD5 with primers containing the relevant sgRNA sequences, followed by Gibson assembly[123].

Cells were transduced with Lenti-Cas9-2A-Blast, or had previously been transduced with an equivalent Hygromycin-resistant vector[124], to stably express Cas9. To generate stable CRISPR/Cas9 deletion mutants, sgRNAs expressed from pSpCas9(BB)-T2A-Puro were transiently transfected into cells as described above, before dilution cloning in flat-bottomed 96-well plates. For other CRISPR/Cas9 experiments, cells were transduced with the indicated sgRNA expressed from pKLV-U6gRNA(BbsI)-PGKpuro2ABFP, pKLV-U6gRNA(BbsI)-PGKblast2ABFP, or tet-pLKO-sgRNA-puro, and assays performed 10–14 days after transduction, following antibiotic selection. All CRISPR/Cas9 experiments were controlled by transduction with an equivalent empty vector.

### Pooled genome-wide CRISPR/Cas9 screening
At least $1.2 \times 10^8$ Cas9-expressing cells of 786O, 786O + *VHL*, RCC4 and RCC4 + *VHL* backgrounds were transduced with the TKOv3 library at a multiplicity of infection of ~30%. Cells were expanded to 15 cm dishes after 27 h, and selected with Puromycin for seven days. Every two or three days, cells were pooled, counted and passaged. After 17 population doublings, genomic DNA was extracted from $4 \times 10^7$ cells using a Puregene Cell Kit (Qiagen Cat#158043). Lentiviral sgRNA inserts were amplified in a two-step PCR using the primers detailed in Supplementary Table 2, with the second step introducing unique barcode sequences for multiplexed sequencing[125]. Amplicons were cleaned with Agencourt AMPure XP beads and sequenced on HiSeq 4000 or NovaSeq 6000 (Illumina) with a custom sequencing primer (Supplementary Table 2). An average of at least 400-fold (786O screen) or 500-

fold (RCC4 screen) representation of each sgRNA within the pool was maintained throughout all stages of the screens.

Sequencing data was processed and analysed using Cutadapt, HISAT2 and Bayesian Analysis of Gene Essentiality version 2 (BAGEL2) in a Snakemake-based bioinformatic pipeline (https://github.com/niekwit/crispr-screens, https://doi.org/10.5281/zenodo.10286661). Read counts for each sgRNA were compared across paired *VHL*-proficient and -deficient cell lines, and between late and early timepoints in each cell line, using BAGEL2, MAGeCK and DrugZ[38–40].

### shRNA-mediated RNA depletion
Oligonucleotides were designed with a TTCAAGAGA hairpin, using shRNA sequences from the Broad Institute RNAi Consortium shRNA Library (Supplementary Table 2). Sequences were cloned into pC.SIREN.Puro by digestion and ligation. Cells were transduced with the indicated vector or a scrambled shRNA control, and assays performed after at least seven days, following Puromycin selection.

### Proliferation and cell death assays
Cells were seeded in six-well plates at predefined densities ($1.5 \times 10^4$ cells/well for 786O cells, $2.5 \times 10^4$ cells/well for other cell lines), and manually counted with a haemocytometer after 96 h. For the IFN-β dosage experiment, cells were seeded with the indicated concentration of IFN-β at $1 \times 10^4$ cells/well in 12-well plates, and final counts after 72 h were normalised to an untreated control. For the Bafilomycin A1 experiment, $5 \times 10^4$ cells were seeded in six-well plates 24 h before treatment with the indicated dose of Bafilomycin A1 or the DMSO vehicle, and counted after a further 48 h.

### Clonogenic assay
Cells were seeded in six-well plates at a density of 100 cells/well and incubated undisturbed for seven days. Colonies were fixed with 100% ice-cold methanol for 10 minutes, visualised with 0.5% crystal violet in 25% methanol, and counted.

### Flow cytometry
Following appropriate staining, cells were collected in 5 ml FACS tubes, centrifuged, washed and resuspended in PBS prior to analysis on a Fortessa flow cytometer (BD Biosciences) using FACSDiva and FlowJo software. Gating strategies are displayed in Supplementary Fig. 9.

Cell death experiments employing SYTOX AADvanced and CellEvent Caspase-3/7 Green were performed according to the manufacturer's instructions, with gates set based on empty vector-transduced or DMSO treated controls. For cell death inhibitor treatments, cells were seeded at a density of $2 \times 10^4$ cells/well in 12-well plates in media containing the indicated drug concentration or a DMSO vehicle. The media was exchanged after 24 h with fresh media containing the same dose of inhibitor, and analysis performed as above after a further 24 h.

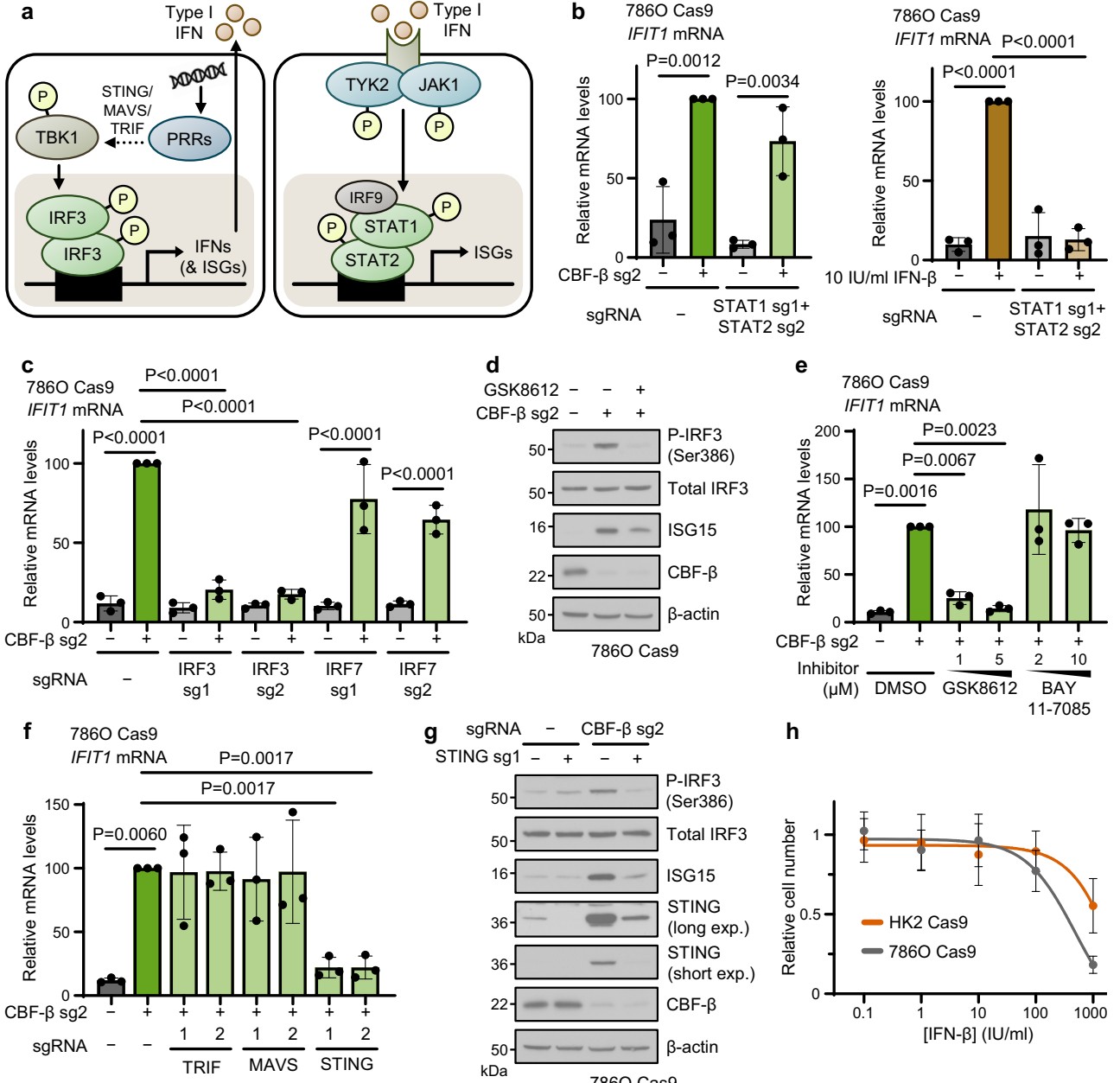

**Fig. 5 | *CBFB* loss induces a STING-TBK1-IRF3 cell-intrinsic ISG response.**
**a** Schematic of the type I IFN signalling pathway. Aberrant or foreign nucleic acids are detected by pattern recognition receptors (PRRs), which signal through adaptor proteins to stimulate TBK1-mediated phosphorylation of IRF3. IRF3 heterodimers translocate to the nucleus to trigger transcription of IFNs and a subset of ISGs. Secreted IFNs act through IFNAR receptors to promote STAT1- and STAT2-mediated ISG expression. **b** qPCR analysis of *IFIT1* expression in 786O Cas9 cells transduced with a vector encoding sgRNAs targeting both STAT1 (sg1) and STAT2 (sg2), or an empty vector control. Cells were additionally transduced with CBF-β sg2, or treated with 10 IU/ml IFN-β for 6 h. $n = 3$ biologically independent replicates. Mean ± SD. Two-way ANOVA. **c** ISG expression is dependent on IRF3. qPCR analysis of cells transduced with sgRNAs targeting CBF-β (sg2), IRF3 and IRF7, or an empty vector control. $n = 3$ biologically independent replicates. Mean ± SD. Two-way ANOVA. **d** Cells were transduced with CBF-β sg2 or an empty vector control, and treated with

5 µM GSK8612 for 24 h (to inhibit TBK1), as indicated. Immunoblot representative of 3 biologically independent replicates. **e** 786O Cas9 cells were transduced with CBF-β sg2 or an empty vector control, treated for 24 h with the indicated doses of the TBK1 inhibitor GSK8612, the NF-κB inhibitor BAY 11-7085, or the DMSO vehicle, and analysed by qPCR. $n = 3$ biologically independent replicates. Mean ± SD. One-way ANOVA. **f** CBF-β loss specifically activates STING. qPCR analysis of cells transduced with CBF-β sg2 alone or in combination with sgRNAs targeting the PRR adaptors TRIF, MAVS and STING, compared to an empty vector-transduced control. $n = 3$ biologically independent replicates. Mean ± SD. One-way ANOVA. **g** Immunoblot of 786O Cas9 cells transduced with sgRNAs targeting CBF-β (sg2) and STING (sg1), or an empty vector control. Representative of 3 biologically independent replicates. **h** Dose-response relationship between IFN-β treatment and cell proliferation over 72 h in 786O Cas9 cells and proximal tubule HK2 Cas9 cells. $n = 6$ biologically independent replicates. Mean ± SD. Source data are provided as a Source Data file.

To assess mitochondrial function, cells were seeded in six-well plates at $8 \times 10^4$ cells/well 24 h prior to analysis with MitoSOX Red, MitoTracker Green FM, and tetramethylrhodamine methyl ester (TMRM). Before staining, control wells were treated with 30 min of Carbonyl cyanide-p-trifluoromethoxyphenylhydrazone (FCCP) or

Antimycin A. For MitoTracker Green and TMRM staining, cells were harvested, centrifuged and stained in FACS tubes for 30 min at 37 °C. MitoSOX Red staining was performed at 37 °C for 10 min directly in six-well plates, having washed cells in Hanks' Balanced Salt Solution (HBSS).

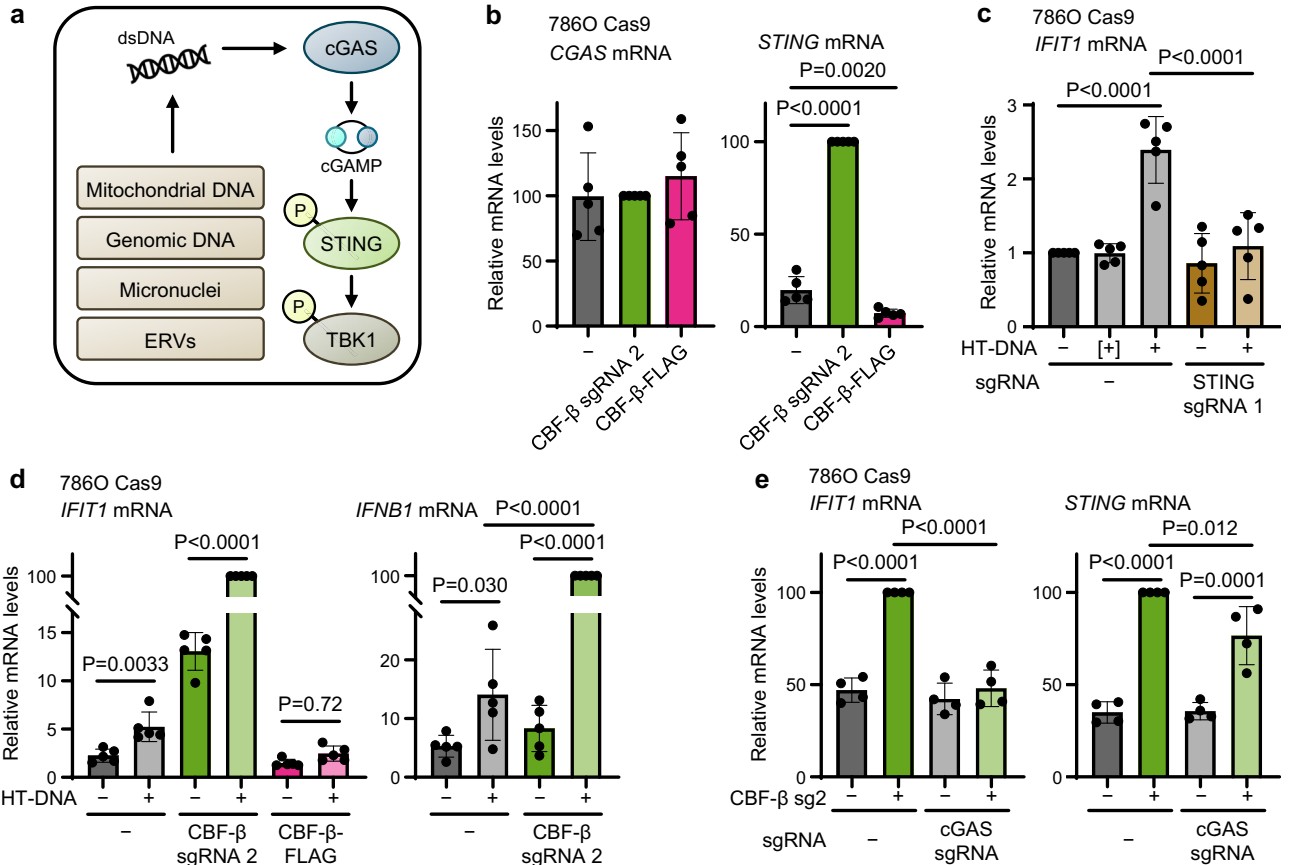

**Fig. 6 | CBF-β represses STING to tune the type I interferon response. a** cGAS-STING signalling axis. cGAS is activated by exogenous or mislocalised double stranded DNA (dsDNA), leading to cGAMP production which triggers a series of phosphorylation events involving STING, TBK1 and IRF3. **b** CBF-β regulates STING transcriptionally. qPCR analysis of *CGAS* and *STING* expression in 786O Cas9 cells transduced with CBF-β sg2, a wild-type CBF-β-FLAG overexpression construct, or an empty lentiviral vector. *n* = 3 biologically independent replicates. Mean ± SD. One-way ANOVA. **c** 786O Cas9 cells were transduced with an sgRNA targeting STING, or an empty vector control, and transfected 6 h prior to analysis with 0.5 µg/ml herring testes DNA (HT-DNA) as indicated. Additional controls (indicated by [+]) were treated for 6 h with 0.5 µg/ml HT-DNA, in the absence of the Lipofectamine 2000 transfection reagent. qPCR analysis of *IFIT1* mRNA expression. *n* = 3 biologically independent replicates. Mean ± SD. Two-way ANOVA. **d** CBF-β controls the magnitude of the ISG response to dsDNA transfection. qPCR analysis of 786O Cas9 cells transduced with CBF-β sg2, a wild-type CBF-β-FLAG overexpression construct, or an empty vector, and either transfected with 0.5 µg/ml HT-DNA for 6 h or left untransfected. *n* = 3 biologically independent replicates. Mean ± SD. Two-way ANOVA. **e** qPCR analysis of 786O Cas9 cells transduced with sgRNAs targeting CBF-β (sg2) and cGAS, or an empty vector. *n* = 4 biologically independent replicates. Mean ± SD. Two-way ANOVA. Source data are provided as a Source Data file.

For intracellular staining of γ-H2A.X, cells were permeabilised with ice-cold 90% methanol for 30 min, incubated at 4 °C for 30 min with the primary antibody in a 1% BSA/PBS incubation buffer, washed in PBS, and stained with the secondary antibody at 4 °C for a further 30 min. Cells were finally incubated in 3 µM propidium iodide/PBS for 15 min before flow cytometry analysis.

**Competitive growth assay**

Seven days post-transduction and following antibiotic selection, $3 \times 10^4$ clonal Cas9-expressing 786O mCherry cells were mixed with an equal number of clonal Cas9-expressing 786O + *VHL* GFP cells in six-well plates. For doxycycline-inducible sgRNA expression vectors, 100 ng/ml doxycycline was applied for nine days following the establishment of the co-culture. At each passage, a subset of each culture was harvested in 3.6% paraformaldehyde for flow cytometry analysis. The 786O mCherry:786O + *VHL* GFP cell ratio was normalised to 0.5 for the first time-point, and then to that of a relevant empty vector or scrambled shRNA control.

**Cell cycle analysis**

BrdU incorporation and propidium iodide staining of total cellular DNA content were used to assay cell cycle phases. For dual staining,

$3.5 \times 10^5$ cells were seeded in 10 cm dishes. After 24 h, cells were incubated in media containing 10 µM BrdU for one hour. Intracellular staining and flow cytometry was performed as described for γ-H2A.X, with an additional treatment step for DNA hydrolysis with 2 M hydrochloric acid for 30 min at room temperature immediately after methanol permeabilisation.

Cell cycle synchronisation was achieved using a double thymidine block method. $3.5 \times 10^4$ cells were seeded in six-well plates. Cells were treated with 2 µM thymidine after seven hours, which was replaced with thymidine-free media after 15 h. After eight hours, media was replaced again with 2 µM thymidine for 17 h. At the end of this second treatment, and for every two hours thereafter for 14 h, samples were harvested in 90% methanol before propidium iodide staining and flow cytometry analysis. An asynchronous control of each experimental condition was maintained in thymidine-free media throughout.

**LDH release cytotoxicity assay**

$5 \times 10^3$ cells were seeded in triplicate in 96-well plates with 100 µl of media. After 24 h, 50 µl supernatant from each well was transferred to a new plate, and the Pierce LDH Cytotoxicity Assay Kit (ThermoFisher Cat#88953) used to assess LDH levels as per the manufacturer's

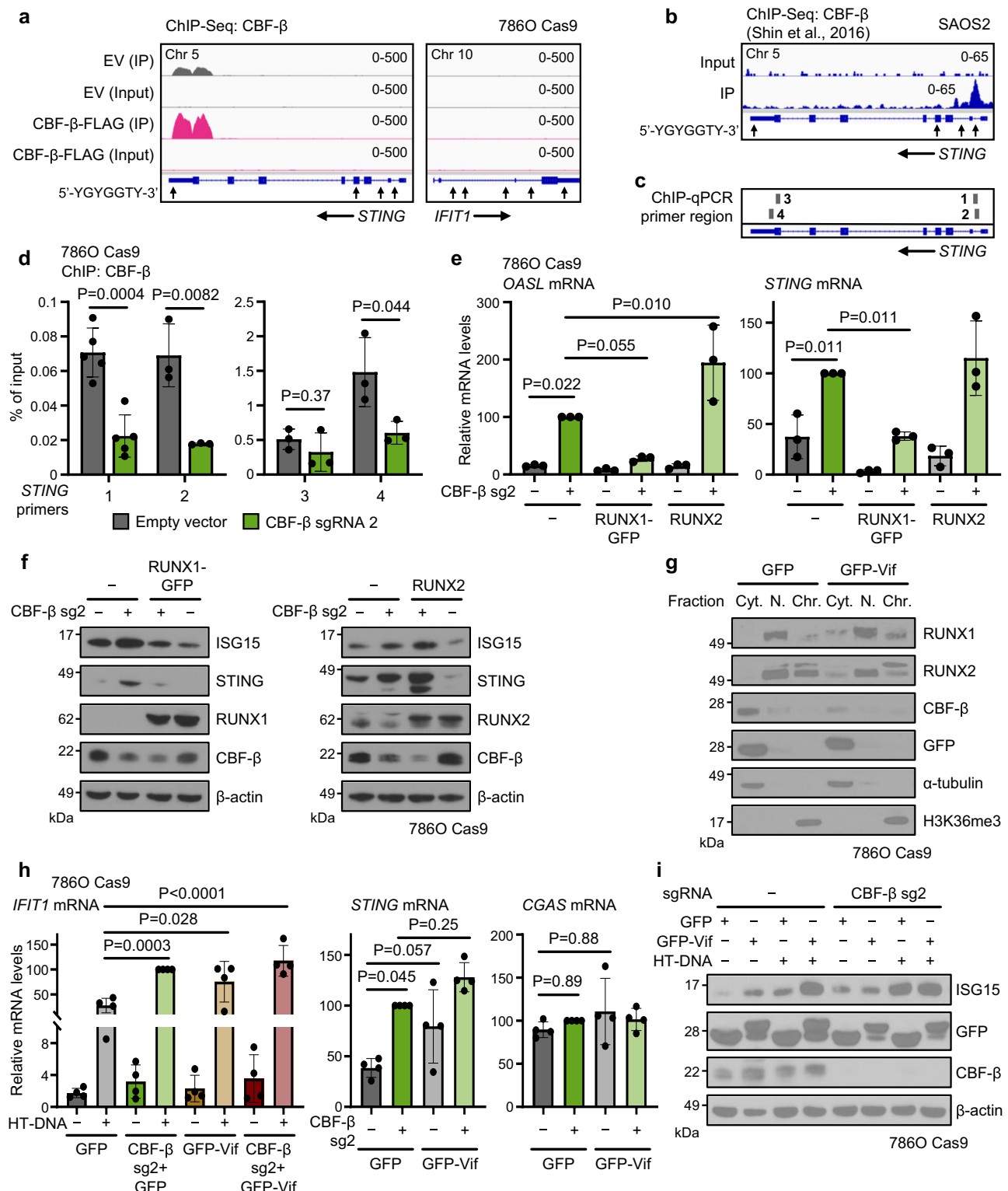

instructions, with absorbance measurements taken using a CLARIOstar Plus plate reader (BMG Labtech). Background absorbance at 680 nm was subtracted from the 490 nm signal, and readings normalised to a cell-free unconditioned media control.

## DNA transfection

$8 \times 10^4$ 786O Cas9 cells were seeded in 2 ml media in six-well plates 24 h prior to transfection with herring testes DNA (HT-DNA). Mixes of 4 µg/ml HT-DNA and 40 µg/ml Lipofectamine 2000, prepared in Opti-MEM I, were combined and gently mixed. After 30 min incubation at room temperature, 500 µl DNA/Lipofectamine 2000 complex was added dropwise to cells.

## Real-time quantitative PCR (qPCR)

Cell lysis was performed in six-well plates at ~70–80% confluence. Total RNA was extracted and purified with the RNeasy Plus Mini Kit (Qiagen Cat#74136), and cDNA generated using Protoscript II Reverse Transcriptase. 20 ng template cDNA was amplified with SYBR Green PCR Master Mix on a QuantStudio 7 Real-Time PCR System (ThermoFisher), using primers specific to the mRNA interest (Supplementary Table 2).

**Fig. 7 | RUNX/CBF-β proteins directly inhibit *STING* transcription. a** Chromatin binding profile of CBF-β at *STING* and *IFIT1*. ChIP-Seq analysis of 786O Cas9 cells expressing the CBF-β-FLAG overexpression plasmid or an empty vector. Mean normalised reads across all replicates displayed using the IGV genome browser. Arrows indicate the directionality of the gene of interest. RUNX consensus motifs (5′-YGYGGTY-3′) are denoted below each track. *n* = 3 (empty vector) or 2 (CBF-β-FLAG) biologically independent replicates. **b** Chromatin binding of CBF-β at the *STING* locus in SAOS2 cells. ChIP-Seq data obtained and reanalysed from Shin et al. (GSE76937)[75], and mean normalised reads across all replicates displayed using the IGV genome browser. *n* = 1. **c, d** Chromatin binding of CBF-β at various sites in the *STING* gene, assayed by ChIP-qPCR analysis of 786O Cas9 cells transduced with CBF-β sg2 or an empty vector (**d**). Primer locations are illustrated in (**c**). Immunoprecipitation of CBF-β was achieved using Cell Signaling Technology antibody Cat# 62184 (primer sets 1 and 2), or Diagenode antibody Cat#C15310002 (primer sets 3 and 4). *n* = 5 (primer set 1) or 3 (primer sets 2, 3 and 4) biologically independent

replicates. Mean ± SD. Unpaired two-sided *t*-test. **e** qPCR analysis of *OASL* and *STING* mRNA expression in 786O Cas9 cells transduced with vectors encoding CBF-β sg2, GFP-tagged RUNX1, RUNX2, or empty vector controls. *n* = 3 biologically independent replicates. Mean ± SD. Two-way ANOVA. **f** Immunoblots of 786O Cas9 cells in the experimental conditions assayed in Fig. 7e. Representative of 3 biologically independent replicates. **g** Subcellular fractionation following transduction of 786O Cas9 cells with an HIV GFP-Vif expression construct, or a GFP-only control. Cyt. cytoplasmic fraction, N. nucleoplasmic fraction, Chr. chromatin-associated fraction. Immunoblot representative of 2 biologically independent replicates. **h, i** Cells were transduced with vectors encoding CBF-β sg2, GFP-Vif, or a GFP-only control, as indicated, and either transfected with 0.5 µg/ml HT-DNA for 6 h or left untransfected. qPCR analysis of *IFIT1*, *CGAS* and *STING* mRNA expression (**h**), and immunoblot analysis of ISG15 levels (**i**). *n* = 4 biologically independent replicates. Mean ± SD. Two-way ANOVA. Source data are provided as a Source Data file.

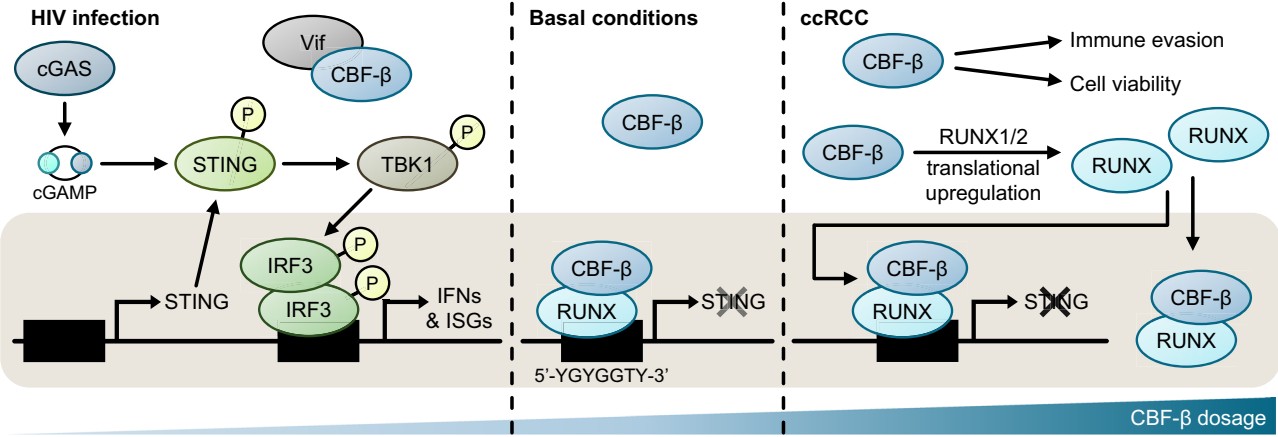

**Fig. 8 | Model of regulation of STING expression by CBF-β/RUNX.** In basal conditions, CBF-β acts as a rheostat to fine-tune cGAS-STING pathway activity. In the absence of CBF-β/RUNX nuclear translocation, as occurs with the sequestration of CBF-β by the lentiviral protein Vif, STING is abundantly expressed, thus amplifying the response to cGAS activation to stimulate signalling through TBK1 and

IRF3, ultimately inducing an IFN/ISG-dependent antiviral state. Conversely, CBF-β levels are elevated in ccRCC, increasing the translation of RUNX1/2 in turn, and permitting cell growth. Additionally, CBF-β and RUNX binding at the *STING* locus suppresses IFN signalling and likely promotes immune escape.

qPCR reactions for each cDNA/primer combination were performed in technical triplicates, and the mean used to calculate the relative transcript abundance with the ΔΔCt method following normalisation to the reference index of a housekeeping gene (*ACTB*).

### RNA sequencing
Total RNA was extracted using the PureLink RNA Mini Kit (Thermo-Fisher Cat#12183025), including on-column DNA digestion with Pure-Link DNase. Library preparation and sequencing were undertaken by GENEWIZ (Azenta), using NEBNext Ultra II RNA Library Preparation Kit (NEB Cat#E7770L) for polyA transcript selection, and NovaSeq 6000 (Illumina). Bioinformatic analysis of RNA-Seq data with DESeq2 was performed as described previously[124], except using Salmon for read mapping and STAR/TEtranscripts for differential expression analysis of transposable elements. Gene Set Enrichment Analysis of Hallmark gene sets was undertaken in R using the fgsea package.

### ChIP-qPCR
786O cells grown to 70% confluence in 15 cm dishes were treated with 1% formaldehyde for 10 min to crosslink proteins to chromatin, and the reaction quenched with 0.125 M glycine for 10 min at room temperature. Cells were washed twice in ice-cold PBS, transferred to tubes, centrifuged at 800 rpm for 10 min and lysed in 500 µl of ChIP lysis buffer (50 mM Tris-HCl pH 8.1, 1% SDS, 10 mM EDTA, and 1×cOmplete EDTA-free protease inhibitor cocktail). Next, samples were incubated for 10 min on ice, and diluted with an equivalent volume of ChIP dilution buffer (20 mM Tris-HCl pH 8.1, 1% (v/v) Triton X-100, 2 mM

EDTA, and 150 mM NaCl) prior to sonication using beads in a Bioruptor (Diagenode) for 20 cycles of 30 s on and 30 s off. Samples were centrifuged at 4 °C for 10 min at 13,000 rpm and supernatants collected and stored at −20 °C as input controls. 200 µl aliquots of the remaining samples were diluted with ChIP dilution buffer to 1 ml, and precleared with 25 µl Protein G magnetic beads at 4 °C for 2 h before overnight immunoprecipitation with the indicated primary antibody at 4 °C. Next, 25 µl Protein G magnetic beads were added and samples incubated for a further two hours at 4 °C. Beads were washed for 5 min with wash buffer 1 (20 mM Tris-HCl pH 8.1, 0.1% (w/v) SDS, 1% (v/v) Triton X-100, 2 mM EDTA, and 150 mM NaCl), wash buffer 2 (as for wash buffer 1 with 500 mM NaCl), wash buffer 3 (10 mM Tris-HCl pH 8.1, 0.25 M LiCl, 7 1% (v/v) NP-40, 1% (w/v) Na deoxycholate, and 1 mM EDTA), and twice with TE buffer (10 mM Tris-HCl pH 8.0, and 1 mM EDTA). Complexes were eluted in 120 µl elution buffer (1% (w/v) SDS and 0.1 M NaHCO₃), and incubated overnight at 65 °C with agitation with 0.2 M NaCl to reverse the formaldehyde crosslinks. Samples were incubated with 20 µg Proteinase K for 4 h at 45 °C, and with RNase H for 30 min at 37 °C to digest protein and RNA, before DNA was purified using the DNA MinElute kit (Qiagen Cat#28206). DNA was analysed by qPCR, and results were normalised to the amplification of the input material for each sample.

### ChIP sequencing
For each replicate, two 15 cm plates of 786O Cas9 cells transduced with a CBF-β-FLAG overexpression vector or an empty vector were fixed and sonicated, as for ChIP-qPCR. 1 ml sonicated sample was diluted

with 4 ml ChIP dilution buffer, transferred to a 15 ml falcon tube, and pre-cleared using 60 µl IgG beads. For ChIP sequencing samples, 30 µl primary CBF-β antibody (Diagenode Cat#C15310002) was used per replicate. ChIPed DNA was eluted in 10 µl elution buffer and submitted for sequencing.

Libraries were prepared using ThruPLEX® DNA-Seq Kit and ThruPLEX DNA Unique Dual Index Kit according to the manufacturer's recommendations. The pooled libraries were sequenced on the Illumina NovaSeq X Plus system (10B flowcell, PE50 read length).

A Snakemake v8.25.5 pipeline was used to analyse ChIP-Seq data (https://doi.org/10.5281/zenodo.138015265): read quality was first assessed with FastQC v0.12.1 and MultiQC v1.21, after which the reads were quality trimmed with TrimGalore. Alignment against the human genome (hg38, build 113) was performed using Bowtie2 v2.5.3. Using SAMtools view v1.20, only fragments with MAPQ > 10 were retained for further analysis. BEDTools intersect v2.31.1 was then used to remove blacklisted regions (obtained from ENCODE, https://www.encodeproject.org/files/ENCFF356LFX). To remove duplicates, fragments were first sorted by coordinate using SAMtools sort, after which Picard MarkDuplicated v3.1.1 was used with the command line flag --REMOVE_DUPLICATES true. The resulting BAM file was indexed using the SAMtools index command. BigWig files for individual replicates were made with deepTools bamCoverage v3.5.5 with the settings binSize 10 and normalizeUsing RPKM. The Principal Component Analysis of these BigWig files was performed with deepTools multiBigWigSummary and deepTools plotPCA, while plotting was done with a custom R script. Peak calling on individual replicates was performed on the deduplicated BAM files with MACS2 callpeak v2.2.9.1 (broad peak calling) with $q = 0.05$ and broad cutoff = 0.05 and the matched input sample as a control. Consensus peaks (BED format) were generated using a custom Python script that identified regions of overlap between all replicates.

To create the coverage plot, mean BigWig files were first generated from all replicates of each condition (IP and input): first a mean Wig file was created using WiggleTools v1.2.11, which was converted to the BigWig format using wigToBigWig v4 (UCSC Genome Browser). To compute the ratio of IP over input, deeptools bigwigCompare was used with the command line flags --operation ratio --skipZeroOverZero. bigWigAverageOverBed v469 (UCSC Genome Browser) was used with the consensus peak file described above to obtain the average ratio of IP over input in the consensus peaks. GNU Awk v5.0.1 was then used to append these scores as an additional column to the consensus peak BED file. Finally, this scored BED file was used as input for the coverage plot using a custom R script that utilised the Bioconductor package ChIPseeker v1.44.0. Genome tracks were visualised with IGV v2.16.0.

## Liquid-chromatography mass spectrometry (LC-MS)

$2 \times 10^6$ cells were harvested in PBS, resuspended in 71 µl resuspension buffer (76 mM triethylammonium bicarbonate (TEAB) pH 8.5, 3 mM MgCl2, 1400 U/ml benzonase, 7 mM tris(2-carboxyethyl)phosphine (TCEP), and 28 mM chloroacetamide), and lysed with the addition of 25 µl 20% SDS for 30 min at 25 °C in the dark. 4 µl 1.25 M Azido-PEG3-Azide was added and samples incubated for 20 min at 55 °C to quench the TCEP. 5 µl aliquots were compared to a standard curve of BSA using a Pierce Microplate BSA assay (ThermoFisher Cat#23252), and samples were normalised to 25 µg by dilution in a 50 mM TEAB, 5% SDS buffer.

To complete the denaturation, a 10% volume of 27.5% phosphoric acid was added to each sample, causing acidification to approximately pH 2. A 6× volume of a wash buffer (100 mM HEPES pH 7.55, 90% methanol) was added, and the solution transferred to an S-trap micro column using a positive pressure manifold (Tecan M10) and in-house fabricated adaptors. Samples were washed with 1:1 (v/v) methanol:chloroform and three times with the wash buffer. S-traps were centrifuged at $4000 \times g$ for 2 min to remove residual wash buffer, and

30 µl of a digestion solution (50 mM TEAB pH 8.5, 0.1% Na deoxycholate) containing 1.25 µg Trypsin/lysC Mix was added. S-traps were incubated for 16 h at 37 °C, and peptides recovered by the addition of 25 µl digestion solution and incubation at room temperature for 15 min, before centrifugation at $4000 \times g$. 40 µl 0.2% formic acid (FA) and 25 µl 50% acetonitrile (ACN) were used in the same manner to elute samples, which were subsequently dried in a vacuum centrifuge equipped with a cold trap. Samples were resuspended in 21 µl 100 mM TEAB pH 8.5. 0.5 µg unique TMTpro labels, resuspended in 9 µl anhydrous ACN, were added to each sample and incubated at room temperature for one hour. To confirm labelling efficiency of >98% and equal loading, 3 µl aliquots of each sample were taken, pooled, and analysed by LC-MS.

Samples were normalised by adjusting for the total reporter ion intensities of the initial test, and were pooled and dried in a vacuum centrifuge. Samples were acidified with ~200 µl 0.1% Trifluoroacetic Acid (TFA), and FA was added until the Na deoxycholate was visibly precipitated. Four volumes of ethyl acetate were added and vortexed vigorously for 10 s. Samples were centrifuged at $15,000 \times g$ for 5 min at room temperature for phase separation, and the aqueous phase transferred to a fresh tube. Samples were partially dried in a vacuum centrifuge and made up to 1 ml with 0.1% TFA. FA was added to achieve a pH <2, and the samples cleaned by solid phase extraction using a 50 mg tC18 Sep-Pak cartridge (Waters) and a positive pressure manifold: the cartridge was wetted with 1 ml methanol followed by 1 ml ACN, equilibrated with 1 ml 0.1% TFA, and each sample loaded slowly. The cartridge was washed with 1 ml 0.1% TFA, and samples were eluted in 750 µl 80% ACN, 0.1% TFA and dried under a vacuum.

Next, samples were resuspended in 40 µl ammonium formate pH 10 and transferred to a glass HPLC vial. Basic pH reverse-phase fractionation was conducted on an Ultimate 3000 UHPLC system (ThermoFisher) equipped with a 2.1 mm × 15 cm, 1.7 µm Kinetex EVO column (Phenomenex) set at a flow rate of 500 µl/minute, using solvent A (3% ACN), solvent B (100% ACN), and solvent C (200 mM ammonium formate pH 10; kept at a constant 10%). Samples were loaded in 90% solvent A for 10 minutes before they were eluted in a gradient of solvent B of 0–10% over 10 min, 10–34% over 21 min, and 34–50% over 5 min, followed by a 10 min wash with 90% solvent B. 100 µl fractions were collected throughout the run, and those containing peptide (determined by ultraviolet absorbance at 280 nm) were recombined across the gradient to preserve orthogonality with on-line low pH reverse-phase separation. Combined samples were dried in a vacuum centrifuge and stored at −20 °C.

Samples were analysed by LC-MS on an Orbitrap Fusion instrument on-line with an Ultimate 3000 RSLC nano UHPLC system (ThermoFisher), using trapping solvent (0.1% TFA), analytical solvent A (0.1% FA) and analytical solvent B (0.1% FA in ACN). After resuspension in 10 µl 1% TFA and 5% DMSO, 5 µl of each sample was loaded onto a 300 µm × 5 mm PepMap cartridge trap (ThermoFisher) at 10 µl/minute for 5 min. Samples were separated on a 75 cm × 75 µm I.D. 2 µm particle size PepMap C18 column (ThermoFisher) with a gradient of analytical solvent B at 3–10% over 10 min, 10–35% over 155 min, and 35–45% over 9 min, followed by washing with 95% analytical solvent B for 5 min and re-equilibration at 3% analytical solvent B. Eluted particles were next introduced to the MS by 2.1 kV electrospray to a 5 cm × 30 µm stainless steel emitter (PepSep). The MS was operated in SPS mode, in which MS1 Scans are obtained in the Orbitrap, CID-MS2 Scans in the Ion Trap and HCD-MS3 acquired in the Orbitrap to resolve reporter ions.

Data were analysed with Peaks 11, and .raw files searched against the SwissProt Human Database with appended common contaminants. The peptide-spectrum match FDR was controlled at 0.1% by decoy database search. Statistical analysis of the relative abundance of identified proteins in each sample was performed with the limma package in R by a moderated $t$-test. A $q$-value was computed to

determine appropriate cut-off values by correcting *P*-values for multiple hypothesis testing with the Benjamini-Hochberg method.

## SDS-PAGE immunoblotting

Cells were lysed at ~70–80% confluence in six-well plates with an SDS loading buffer (1% SDS, 50 mM Tris pH 7.4, 150 mM NaCl, 10% glycerol, and 2 μl/ml benzonase) and incubated on ice for 20 min before heating at 90 °C for 5 min. Proteins were resolved by SDS-PAGE (Bio-Rad), and transferred to methanol-activated polyvinylidenedifluoride (PVDF) membranes. Membranes were blocked with 0.2% TWEEN-20/PBS containing 5% skimmed milk powder and 1% BSA, probed with the appropriate primary and secondary antibodies for at least one hour each, and developed with Pierce Enhanced Chemiluminescent, SuperSignal West Pico Plus Chemiluminescent, or SuperSignal West Dura Extended Duration substrates. PBS was substituted with TBS for the detection of phosphorylated proteins. β-Actin levels were used to confirm the equivalent loading of each sample. Indicated protein sizes were determined by comparison to a SeeBlue Plus2 pre-stained protein standard.

## Immunoprecipitation

786O cells were lysed in 1 ml 1% Triton X-100/TBS supplemented with 1×cOmplete EDTA-free protease inhibitor cocktail at 4 °C for 30 min. Lysates were centrifuged at $14,000 \times g$ for 15 min at 4 °C, and the supernatant collected. 20 μl of each sample was taken in an equal volume 2×SDS loading buffer as an input, and the remainder precleared with Pierce Protein G magnetic beads for one hour at 4 °C. Supernatants were incubated overnight with 2 μl of the indicated primary antibody at 4 °C with rotation. Next, Pierce Protein G magnetic beads were added for two hours, and samples were washed three times in 1% Triton X-100/TBS lysis buffer. Bound proteins were eluted in 2×SDS loading buffer and analysed by SDS-PAGE immunoblotting.

## Subcellular fractionation for immunoblotting

$2 \times 10^6$ cells were lysed and incubated with rotation in buffer A (0.1% IGEPAL CA-630, 10 mM HEPES, 1.5 mM MgCl$_2$, 10 mM KCl, 0.5 mM DTT, and 1×cOmplete EDTA-free protease inhibitor cocktail) for 10 min at 4 °C before centrifugation at $1400 \times g$ for 4 min at 4 °C. The supernatant, representing the cytoplasmic fraction, was collected in 2×SDS loading buffer. The nuclear pellet was washed in buffer A, with centrifugation as before, and resuspended in buffer B (20 mM HEPES, 1.5 mM MgCl$_2$, 300 mM NaCl, 0.5 mM DTT, 25% glycerol, 0.2 mM EDTA, and 1×cOmplete EDTA-free protease inhibitor cocktail) for 10 minutes at 4 °C. Samples were centrifuged at $1700 \times g$ for 4 min at 4 °C, and the soluble nucleoplasmic fraction and insoluble chromatin pellet were each collected and resuspended in 2×SDS loading buffer. Subcellular fractions were analysed by SDS-PAGE immunoblotting.

## Mitochondrial DNA leakage assay

$1 \times 10^7$ cells were harvested by trypsinisation, pelleted and washed in PBS. Pellets were resuspended in 550 μl ice-cold Cytosolic Extraction Buffer (NaCl 150 mM, HEPES 50 mM, digitonin 200 μg/ml, hexylene glycol 1 M, and 1×cOmplete EDTA-free protease inhibitor cocktail) and incubated on ice for 10 min. Samples were centrifuged at $2000 \times g$ for 10 min at 4 °C. The supernatant (containing the cytosolic fraction) was transferred to a fresh tube, and the pellet (containing the reference nuclear fraction) resuspended in 550 μl RIPA lysis buffer (25 mM Tris-HCl pH 7.6, 150 mM NaCl, 1% NP-40, 0.1% Na deoxycholate, 0.1% SDS, and 1×cOmplete EDTA-free protease inhibitor cocktail). Cytosolic samples were incubated for a further 10 min on ice, centrifuged as before, and the supernatant isolated. Next, samples were incubated with 500 μg/ml RNase A for 1 h at 37 °C, and then with 10 mg/ml Proteinase K for a further 1 h at 55 °C. DNA was isolated by phenol-choloroform isoamyl alcohol extraction by adding 500 μl UltraPure™ Phenol:Chloroform:Isoamyl Alcohol (25:24:1, v/v), shaking vigorously

for 20–30 s, and centrifugation at 18,300 for 15 min at 4 °C. DNA was precipitated by transferring 400 μl of the supernatant to a fresh tube containing 500 μl 100% isopropanol. Samples were vortexed and centrifuged at $18,300 \times g$ for 10 min at 4 °C, and the supernatant discarded. The pellet was washed with 500 μl 70% ethanol, the supernatant removed and the pellet dried at 24 °C. Finally, the pellet was resuspended in 200 μl 10 mM Tris-HCl pH 8.0, and 10 ng of DNA used for qPCR analysis of mitochondrial genes. Cytoplasmic mtDNA abundance was normalised to the amount of *ACTB* DNA in the nuclear fraction for each sample.

## Bioenergetic analysis

Oxygen consumption rates (OCRs) were measured using a Seahorse XF analyser in order to calculate basal respiration. The Mito Stress Test was performed according to the manufacturer's instructions using Seahorse XF FluxPak consumables (Agilent Technologies). $1.2 \times 10^4$ cells were plated in FluxPak 96-well plates and incubated at 37 °C for 24 h. The medium was replaced with Seahorse XF RPMI, pH 7.4 supplemented with 1 mM sodium pyruvate, 2 mM L-glutamine, and 10 mM glucose, and cells were maintained at 37 °C in a non-CO$_2$ incubator for 1 hour before the run. Analysis was performed using the program settings: mix for 3 min and measure for 3 min ×3; inject 1.5 μM oligomycin; mix for 3 min and measure for 3 min ×3; inject 1 μM FCCP; mix for 3 min and measure for 3 min ×3; inject 1 μM Rotenone, 1 μM Antimycin A and 2.5 μM Hoechst 33342; mix for 3 min and measure for 3 min ×3. Basal respiration was calculated by subtracting the OCR following Rotenone and Antimycin A treatment from the basal OCR. Following the assay, results were normalised for cell number by quantifying Hoescht 33342 staining using a CLARIOstar Plate Reader at 355-20/455-30 nm.

## Immunofluorescence

$5 \times 10^4$ 786O cells were seeded on FCS-precoated coverslips in 24-well plates. 10 μM Camptothecin was added to a control well after 24 h, and after a further 24 h, all coverslips were washed twice in Dulbecco's PBS (D-PBS) and fixed with 4% paraformaldehyde for 15 min. Cells were washed twice in D-PBS, permeabilised in 0.1% Triton X-100 for 10 min, washed again, and blocked in 4% FCS in D-PBS. The coverslips were stained overnight with a 1:400 dilution of the anti-Phospho-Histone 2A.X (Ser139) primary antibody at 4 °C, then washed five times with D-PBS and counterstained with a 1:800 dilution of the Alexa-Fluor 488 anti-Rabbit secondary antibody for 90 min at room temperature. Cells were washed five times in D-PBS and mounted in the glycerol-based mountant AF1 plus DAPI prior to imaging with a Zeiss LSM 980 with Airyscan confocal microscope. Foci were counted in a blinded and unbiased manner using Fiji (ImageJ) software.

## Analysis of TCGA and CPTAC expression and survival data

*CBFB* mRNA expression and survival data for ccRCC tumours were obtained from The Cancer Genome Atlas (TCGA), and CBF-β protein levels and survival data obtained from the Clinical Proteomic Tumor Analysis Consortium (CPTAC)[7,126,127]. CBF-β protein abundance was plotted for tissues from ccRCC patients for whom analysed matched normal kidney tissue was available. The R package survminer was used to perform log-rank tests and plot Kaplan-Meier curves.

## Single-cell expression of interferon pathway genes

Single-cell data was downloaded from Li et al. (https://data.mendeley.com/datasets/g67bkbnhhg/1)[48]. The anndata file was first converted to a Seurat object using the sceasy package in R. The single-cell expression data were normalised and variance stabilised using regularised negative binomial regression using the SCTransform function from the seurat R package. Cell types annotated as either 'Low quality' or 'Unknown' were removed with the remaining cells collated into their broad principal lineages of epithelial cells, RCC cells, NK cells, myeloid

cells, fibroblasts, endothelial cells, T cells and B cells. The average normalised expression data for these cell types in relation to interferon and STING pathway genes were plotted using the DotPlot function in R.

## Tumour xenografts

Animal experiments were performed according to protocols approved by either the University of Cambridge Animal Welfare and Ethical Review Board in compliance with the Animals (Scientific Procedures) Act 1986 and UK Home Office regulations, or by the Institutional Animal Care and Use Committee of UT Southwestern Medical Center following NIH guidelines. The maximal tumour size and burden set by the relevant ethics committee was a tumour area of $1.2\,cm^2$ for subcutaneous xenografts, and for orthotopic xenografts, humane endpoint criteria were used, including total body weight loss of greater than 15–20%, impaired mobility, abdominal distention, and other indications of compromised animal welfare; no animals exceeded these limits.

Female NSG mice (NOD.Cg-$Prkdc^{scid}$ $Il2rg^{tm1Wjl}$/SzJ (Charles River RRID:IMSR_JAX:005557)) used for subcutaneous xenograft were housed at a density of five animals per individually ventilated cage with *ad libitum* access to food and water in a specific pathogen-free unit. 786O Cas9 cells expressing the indicated sgRNAs were harvested and washed in D-PBS, and $1 \times 10^7$ cells were administered via subcutaneous injection to the dorsal flank of 8-week-old NSG mice in a blinded and randomised manner between cages. Tumour volume was determined using callipers as the product of $0.5 \times$ maximum length$^2 \times$ maximum breadth, and mice were euthanised at 28 days.

Orthotopic xenografts of 786O Cas9 cells stably expressing luciferase and doxycycline-inducible sgRNA vectors were performed in 6–8 week old male and female NSG mice (UTSW Animal Resource Center) as described previously[46]. $1 \times 10^6$ viable cells were resuspended in 20 µL PBS containing 50% Matrigel and injected orthotopically into the left kidney of each NSG mouse. Following successful implantation, mice were fed Purina rodent chow #5001 with 2000 ppm doxycycline to induce sgRNA expression. Tumour growth was monitored weekly by bioluminescence imaging using the SPECTRAL AMI-HTX system. Mice were euthanised after a further eight weeks, and the tumour mass was calculated by subtracting the weight of each right kidney from the corresponding left kidney. Metastasis was evaluated in isolated lungs by bioluminescence.

## Quantification and statistical analysis

Quantitative data are expressed as the mean of biological repeats ±1 standard deviation (SD) or ±1 standard error of the mean (SEM). *P*-values were calculated using two-tailed Student's *t* tests, two-tailed Mann–Whitney U tests, or analysis of variance (ANOVA), as indicated in figure legends. Statistical analyses of CRISPR/Cas9 screens, RNA-Seq, mass spectrometry, and TCGA and CPTAC data are described in the relevant method sections. The number of biologically independent repeats (independent transductions for experiments involving cell culture, or number of mice for xenograft experiments) is specified in figure legends. Data presented from CRISPR/Cas9 screens are derived from a single replicate in each cell line. Figures were prepared and statistical analyses performed using GraphPad Prism or R. Software used in this study is indicated in Supplementary Table 1.

## Reporting summary

Further information on research design is available in the Nature Portfolio Reporting Summary linked to this article.

## Data availability

Raw data from RNA sequencing, ChIP sequencing and CRISPR/Cas9 screens have been deposited at GEO and are publicly available at GSE300828, GSE270775, and GSE270776. Raw mass spectrometry data have been deposited at ProteomeXchange and are publicly available at PXD074426. Source data are provided with this paper.

## Code availability

Code for the CRISPR screen analysis pipeline is available at https://github.com/niekwit/crispr-screens (https://doi.org/10.5281/zenodo.10286661), the differential transcript analysis pipeline with Salmon/DESeq2 at https://github.com/niekwit/rna-seq-salmon-deseq2 (https://doi.org/10.5281/zenodo.10139567), TE analysis pipeline at https://github.com/niekwit/rna-seq-star-tetranscripts (https://doi.org/10.5281/zenodo.10027278), and ChIP-Seq analysis pipeline at https://doi.org/10.5281/zenodo.138015265.

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

## Acknowledgements

We thank all members of the Nathan lab for their helpful comments on the work and manuscript. We also thank Jan Rehwinkel for discussions regarding interferon signalling. This work was supported by a Wellcome Senior Clinical Research Fellowship to J.A.N. (215477/Z/19/Z), a CRUK PhD studentship to J.A.C.B., and a Lister Institute Research Fellowship to J.A.N. The Q.Z. lab was supported by Cancer Prevention and Research Institute of Texas (CPRIT) (RR190058) and V.H.L. synthetic lethality-related research in the Q.Z. lab is partially supported by National Cancer Institute (R01CA284591). A.O.S. was supported by a grant from the Wellcome Sanger Institute and Wellcome Trust (206194) and J.J.S. was supported by a grant from Open Targets (OTAR2080). T.J.M. is supported by a CRUK fellowship (C63474/A27176). N.J.M. is supported by the Evelyn Trust ref 24/55: (Med-24-2316). This research was supported by the National Institute for Health and Care Research (NIHR) Cambridge Biomedical Research Centre (NIHR203312). The views expressed are those of the authors and not necessarily those of the NIHR or the Department of Health and Social Care.

## Author contributions

Conceptualization, J.A.C.B. and J.A.N.; Methodology, J.A.C.B., T.P., J.C.W., N.W., Q.Z., N.J.M., B.M.O., A.O.S. and J.A.N.; Investigation, J.A.C.B., T.P., Q.L., N.W., J.C.W., J.J.S., T.J.M., A.O.S., Q.Z. and J.A.N.; Writing – original draft, J.A.C.B., T.P. and J.A.N.; Writing – reviewing and editing, all authors; Funding acquisition, J.A.N.; Resources, N.J.M., A.O.S., Q.Z and J.A.N.; Supervision, B.M.O., A.O.S., Q.Z. and J.A.N.

## Competing interests

The authors declare no competing interests.
