## [Transparent Peer Review file · Nature Communications]

VHL synthetic lethality screens uncover CBF- β as a negative regulator of STING

Corresponding Author: Professor James Nathan

Version 0:

Reviewer comments:

Reviewer #1

(Remarks to the Author)

Berlin et al describe a series of genetic screens ostensibly to discover genes that are essential in the context of VHL loss, the signal genomic lesion in kidney cancer (clear cell renal cell carcinoma, ccRCC). The authors perform genome-scale CRISPR knockout screens in two ccRCC cell lines, 786O and RCC4, which carry the signature homozygous VHL loss, and again in those same cells with exogenously expressed VHL.

It is not clear how well-controlled the CRISPR screens are. The authors use the TKOv3 one-component library. Some quality control figures are present in Supp Fig 1, but the relative paucity of hits (genes with Bayes Factor > 0 or some higher threshold) shown in Fig 1C suggests that quality was not optimal. It would be useful to have a precision-recall curve, using common reference sets of essential and nonessential genes (e.g. for essentials, Hart et al, G3, 2017 – the Hart et al, Cell 2015 set is not optimal – and for nonessentials, Hart et al MSB 2014). Other data, such as cell-line specific nonexpressed genes, may be substituted for nonessentials without loss of information. Comparison of the PR curves of these screens to those publicly available ccRCC genome-scale cell line screens in DepMap would provide further evidence of screen quality. This information is important, as it gives some insight into the false negative rate of a cell line screen, which is critical in evaluating whether a hit found in one screen but not another reflects underlying biology or just a missed call in the other screen.

To identify genes providing proliferative advantage (e.g. KEAP1 in this case), a better approach is described in Lenoir et al, Nat Comms 2021. This is far better than Bagel scores, although the outlier nature of KEAP1 knockout is a strong signal here.

The cherrypicking of target genes (Fig 1d) is useful for narrowing the reader's attention to the genes of interest but the display is not convincing. It is standard to calculate and display fold change of normalized end point cells relative to some early point (plasmid or T0), rather than read counts, alongside some negative control (again, nonessentials are useful here) whose distribution is centered around zero.

For comparing differential essentials between cell lines, the approach is flawed. The difference in proliferation rates between the VHL+ and VHL- cells will confound a competition assay for single gene effects. A better approach would be to use an algorithm such as DrugZ (Colic et al, Genome Med 2019; see, e.g., Karasu et al Nat Biotech 2024 for a recent example of its application) to compare endpoint readcounts of screens in VHL+ and VHL- cells.

My motivation for stressing controls so firmly is that the main gene chosen for follow up, CBF β , is a very weak hit in Fig 1d. More importantly, it is not a hit in DepMap. In fact CBF β knockout shows strong phenotype in leukemia, lymphoma, and myeloma cells (Chronos score < -0.5), but is never essential in any of the >25 kidney cancer cell lines screened by DepMap. The in vivo data is compelling, but is totally inconsistent with the in vitro data that were ostensibly used to nominate CBF β for validation.

The authors should either perform a more realistic analysis of their screen data, or find another way to get to the CBF β hypothesis in ccRCC.

Reviewer #2

(Remarks to the Author)

In this study, the authors performed synthetic lethal screens and identified CBFb deletion as synthetic lethal to VHL loss. Mechanistically, they propose that CBFb negatively regulates STING. Overall, the finding that CBFb and VHL are synthetic lethal is potentially interesting. However, the mechanism underlying the role of CBFb in this synthetic lethality remains unclear, and some of the data do not support the model and appear to be conflicting. Additionally, the manuscript's writing needs improvement, as some data contradict the conclusions presented in the figures. The primary concerns are listed below. If the authors can sufficiently address these concerns, I am willing to review a revised version of the manuscript.

Concerns:

1. A major issue about the study is that the results have not provided reasonable mechanistic explanation for the synthetic lethality of CBFb and VHL loss, the major topic of this study. Type I interferon activation induces a plethora of phenotypes in cells, including senescence, apoptosis, cell cycle arrest, etc. Why does CBFb deletion-induced type I IFN activation selectively kill VHL-null ccRCC? Without the mechanism, the main conclusion is not well supported.
2. The increase in STING protein alone is not sufficient to activate the type I interferon pathway. Therefore, CBFb deletion must activate an upstream regulator of STING. The authors should employ a candidate approach to knock out potential upstream activators and elucidate the mechanism of type I IFN activation. The strategy outlined in the Cell paper, "Inhibiting DNA Methylation Causes an Interferon Response in Cancer via dsRNA Including Endogenous Retroviruses," may be helpful to the authors (ScienceDirect, Volume 162, Issue 5, 27 August 2015, Pages 974-986).
3. The data suggest that cGAS, an upstream activator of STING, is required for type I IFN activation upon CBFb depletion (Figure 6e). As cGAS is a double-stranded DNA (dsDNA) sensor, the authors should conduct experiments to identify the source of dsDNA.
4. The whole manuscript is about ccRCC, and the GFP-Vif section in Figure 7 is irrelevant to the study. While it is acceptable if the authors aim to use GFP-Vif to degrade CBFb, this approach does not add significant value beyond the use of sgRNA or shRNA for CBFb. It remains unclear why the authors conducted an HIV-related study and how they concluded that the CBFb amount is lower than the basal conditions (as shown in the bottom of Figure 7).
5. It is puzzling that the induction of ISGs does not require STAT1, STAT2, and IRF9 (Extended Data Fig. 4a). On page 7, first paragraph, it states, "To our surprise, however, knockout of STAT1, STAT2, or IRF9 individually, or in combination, did not prevent IFIT1 induction (Fig. 5b and Extended Data Fig. 4a-c)." One possibility is that the gRNAs for STAT1, STAT2, and IRF9 generated a polyclonal pool, with some cells having reduced STAT1, STAT2, or IRF9 levels, and others remaining wild type. This polyclonal nature could result in the negative outcome, particularly in the context of the autocrine and paracrine effects of IFN- β 1. The authors should select at least two single clones for STAT1, STAT2, and IRF9 knockout and repeat the experiment in Extended Data Fig. 4a.
6. For Extended Data Fig. 4, the title is "ISG transcription following CBFb deletion is mediated by the STING-TBK1-IRF3 axis." However, the data appear to suggest otherwise. Please clarify this discrepancy.
7. In Figure 4e, CBFb N104A rescues the effect of IFIT1 similarly to WT, suggesting that binding to RUNX proteins is not necessary for the repression of type I interferon. This result contradicts the model proposing that the CBFb-RUNX complex suppresses STING. It has been established that CBFb enhances the chromatin binding of RUNX proteins. Therefore, reconciling this result with the proposed model requires a logical explanation.
8. In Figure 6f, the classical ChIP analysis has several issues. Firstly, it was heavily normalized. The authors should present the percentage of input on the Y-axis. Secondly, ChIP-seq or CUT&RUN analyses of CBFb or RUNX2 should be performed to unbiasedly detect the binding of CBFb and RUNX2 genome-wide.
9. The claim that RUNX1 directly binds to the promoter region of STING is questionable because the ChIP-seq data were obtained from a public dataset involving RUNX1 in K562 cells (chronic myeloid leukemia). Transcriptional and epigenetic features are cell-line specific. Thus, binding of RUNX1 in K562 cells does not necessarily indicate binding in ccRCC cell lines. The authors should perform ChIP-seq analysis of RUNX1 in ccRCC cell lines to substantiate this claim.
10. The authors claim that IRF3 directly induces ISGs, which contradicts existing literature. Therefore, robust data are needed to support this claim. IRF3 ChIP-seq data is required to substantiate the assertion. The authors have only performed an IRF3 knockout and demonstrated that IRF3 is necessary, which this piece of data does not indicate IRF3 directly induce ISGs.

Reviewer #3

(Remarks to the Author)

This study offers a novel and interesting insight into the mechanism of "synthetic lethality" in VHL-null kidney cancers. In this manuscript, the authors utilized a genome-wide CRISPR knockout screen to identify the synthetic lethality interactors of von Hippel-Lindau (VHL) in clear cell renal cell carcinoma (ccRCC). They found that the loss of CBF- β suppresses the growth of VHL-null tumors both in vitro and in vivo. Mechanistically, the authors demonstrated that the CBF- β /RUNX complex binds to the STING promoter, inhibiting STING transcription. The loss of CBF- β leads to cell-intrinsic STING activation, resulting in the upregulation of type I interferon and the expression of interferon-stimulated genes (ISGs), which ultimately arrests the cancer cell cycle.

This research demonstrated innovation and provided a potential therapeutic target for ccRCC. However, more robust and convincing evidence is required to support the hypothesis.

Major comments :

1. Previous studies (Refs. 24-34 mentioned in the manuscript) have explored various 'synthetic lethality' interactors of VHL. So, what is the advantage of this study or the new target of CBF-b. Whether other synthetic lethality interactors of VHL (eg. ROCK1, J M Thompson, et. al, Oncogene, 2017) also have similar functions in the ccRCC case? What is the specificity of CBF-b as the 'synthetic lethality' interactors of VHL in the context of ccRCC?
2. Initially, the author aimed to find the potential therapeutic targets for ccRCC and screened out CBF-b as a 'synthetic

lethality' interactor of VHL. Can the author target CBF-b/RUNX-STING-IRF3-IFN axis for the treatment of ccRCC in a mouse model?

3. In the manuscript, the author focuses solely on tumor cells. If the loss of CBF-b activates the STING signaling pathway in ccRCC, how does this affect other immune cells in the tumor microenvironment (TME)? Additionally, what impact could this have on T cells, natural killer (NK) cells, or other immune cell populations in systemic conditions?

4. The author claimed that the loss of CBF-b in VHL-null 786O cells activates STING-IFN-ISG signaling. If the 786O cells produce and secrete type I interferon (IFN), why does the secreted IFN not affect the 786O+VHL cells in the co-culture experiment (Fig. 1e)? Could it be that STING activation triggers type I IFN-independent signaling pathways, such as autophagy and autophagy-mediated cell death?

5. In Fig 1a, VHL was reconstituted in the 786O and RCC4 cell lines. Why do the Western blot bands differ in these two cell lines?

6. In Fig. 5g, why is the STING protein absent in the 786O Cas9 (sgCtrl) cells?

7. In Ex.Fig.6b and 6c, the knockout of CBF-b in RCC4 cells increases the mRNA level of cGAS and STING. However, in RCC10 cells, the knockout of CBF-b appears to have no effect on the regulation of either cGAS or STING levels. This suggests that the idea that CBF-b/RUNX binds to the STING promoter region to repress STING transcription may not be a universal model.

8. Why did the author use the ChIP-Sequencing data from the K562 cell line instead of the ccRCC?

9. In Fig. 6f, ChIP analysis using the CBF-b antibody showed that the levels of STING primers 1 and 3 were reduced in CBF-b KO cells. However, when using the RUNX2 antibody, only primer 1 exhibited a reduction, while primer 3 did not.

Why? Does this suggest that primer 1 is more significant? Additionally, why is the sample size for primers 1 and 2 listed as n=5, while for primers 3 and 4, it is n=3?

Minor point :

1. Some of the western blot bands are too faint. For example, in Fig 4e, the band of IRF7 is barely visible, especially for the sg control.

Reviewer #4

(Remarks to the Author)

The manuscript by Bertlin et al. identifies the CBF-B transcriptional regulator as a synthetic lethal partner of VHL loss in kidney cancer cells. In this study, performed a pooled genome wide CRISPR/Cas9 screen to identify synthetic lethal partners of VHL loss in 2 kidney cancer cell lines. They demonstrate that loss of CBF-B selectively reduces the growth and survival of VHL deficient kidney cancer cell lines. Importantly, genetic inhibition of CBF-B reduces kidney tumor growth in vivo. Mechanistically, this study identifies CBF-B as a potential therapeutic target for the treatment of kidney cancer and identifies novel biological insights into the regulation of the STING pathway in kidney cancer. However, the manuscript requires further clarification and revision before consideration for publication. Please see the detailed comments below.

Figure 1- CBF-B was identified as a synthetic lethal partner of VHL loss in kidney cancer cells. Why is CBF-B loss only lethal only in VHL null cells? The expression levels appear to be similar, but the transcriptional activity appears to be different in the VHL proficient deficient setting.

Figure 2a- The Kaplan-Meier curve showing CBF-B mRNA expression correlates with poor survival in the KIRC cohort would be nicely complemented with data showing if CBF-B protein levels are either increased in VHL deficient kidney cancer compared to normal kidney tissue and/or if CBF-B protein levels correlate with patient outcomes.

Figure 2g- The kinetics of the subcutaneous tumor growth curves would be important to show here.

Figure 3- Does rescue of RUNX expression restore growth and survival in the CBF-B knockout cells? This would be important to determine if RUNX proteins are functionally involved in the CBF-B growth and survival phenotype.

Figure 3-4- Is it thought that RUNX is also involved in the CBF-B regulation of the ISG? It would be helpful to link the RUNX story to the STING story with data if they are proposed to be linked, or are they separate mechanisms?

Figure 5h- A better comparison here would be to compare the sensitivity of the 786O to the 786O+VHL cells. Are the VHL deficient cells more sensitive to IFN- β mediated cell death?

Figure 5 extended data- Does CBF-B loss in VHL deficient kidney cancer cells result in increased numbers of γ -H2AX foci formation and DNA damage? The number of foci rather than total signal is important to rule out that CBF-B loss does not induce DNA damage.

Discussion- Can you please comment on the therapeutic potential of targeting CBF-B for cancer therapy. Are there current inhibitors, what is the potential toxicity of systemic inhibition?

Version 1:

Reviewer comments:

Reviewer #2

(Remarks to the Author)

In this revision, the authors have incorporated new data in an effort to address several key points, and the manuscript reflects some improvement over the previous version. However, substantial concerns regarding the proposed model remain unaddressed, and issues related to data interpretation and internal inconsistencies persist. The following points outline these unresolved concerns:

Overview of the main claims in this revision

The revised manuscript presents a model summarized in Figure 8, which includes several major claims:

Claim 1: CRISPR screening reveals that depletion of CBF- β is synthetically lethal in the context of VHL loss in clear cell renal cell carcinoma (ccRCC).

Claim 2: Loss of VHL leads to increased levels (or dosage) of CBF- β (as shown by comparing the middle and right panels in Figure 8), which is proposed as a partial mechanism underlying the synthetic lethality.

Claim 3: CBF- β directly represses STING, which modulates the type I interferon pathway. CBF- β binds directly to the STING locus, and although RUNXs are involved, the interaction between CBF- β and RUNXs is not required for this repression.

Claim 4: The authors propose that the STING–TBK1–IRF3 axis directly regulates the expression of interferon-stimulated genes (ISGs). They argue that STAT1/STAT2 are not involved in ISG expression; instead, IRF3 directly induces ISGs.

This reviewer has no major concerns regarding Claim 1, which is supported by the screening results and validation data showing that CBF- β depletion preferentially kills VHL-null cells.

Major Concerns

1. Claim 2 — that loss of VHL increases the dosage of CBF- β (Fig. 8)— appears weak and is contradicted by the data. It seems the authors propose this mechanism to partially explain the observed synthetic lethality. However, if that is the case, the data presented in Extended Data Figures 1d, 2d, and 3c do not support the claim. In fact, CBF- β levels are slightly lower in VHL-null cells compared to VHL-reconstituted cells, which directly contradicts the proposed model.

2. Claim 3 — that CBF- β directly represses STING mRNA, and although RUNXs are involved, the interaction between CBF- β and RUNXs is not required for this repression — lacks logical coherence and is not adequately supported by the data. In Figure 3, the authors attempt to demonstrate that the interaction between CBF- β and RUNXs is not required for ISG induction (Figures 3e, 3f, and 3g), yet both CBF- β and RUNXs are independently involved. The key unresolved question is: how can this be reconciled?

The authors propose several possibilities. For instance, if CBF- β merely regulates RUNX translation, as suggested in their cited work (Malik, 2019), and RUNX proteins can repress STING and ISGs independently of CBF- β , then the model shown in Figure 8 is inaccurate or unsupported — it depicts RUNXs recruiting CBF- β to the STING locus. Given that CBF- β is an indispensable binding partners of RUNX, this possibility is highly unlikely, if not impossible.

Moreover, in this revision, the authors claim that CBF- β directly binds to the gene body of STING. However, since CBF- β lacks a DNA-binding domain, it remains unclear how it is recruited to this site since the authors now try to claim that the interaction between CBF- β and RUNXs is dispensable for the regulation. Do RUNXs bind to the STING gene body in ccRCC cells? If so, why is their interaction with CBF- β not required?

As it stands, Claim 3 is not convincingly supported by the data presented.

3. Claim 4 — that IRF3 directly induces ISGs and that STAT1/STAT2 are not involved — requires more robust experimental support. To demonstrate direct regulation by IRF3, the authors performed conventional ChIP assays (Extended Data Fig. 5g). However, the signal (% of input) was low and nearly indistinguishable from background noise. Among the three genes tested, only IFIT2 showed a marginal yet statistically significant enrichment. Importantly, IFIT1 — a central model gene used throughout the manuscript — showed no detectable difference, raising doubts about whether IRF3 directly regulates IFIT1 or other ISGs.

At present, the data only establish that IRF3 is required for ISG expression, without clarifying the underlying mechanism. To substantiate the claim of direct regulation, the authors should perform IRF3 ChIP-seq, which would offer a more comprehensive and convincing assessment of IRF3 binding across ISG loci.

In addition, it is worth noting that STING has been shown to activate noncanonical NF- κ B signaling, which promotes cell survival and metastasis (see Bakhomou et al., Nature: "Chromosomal instability drives metastasis through a cytosolic DNA response", 2018). This alternative pathway should be considered in the interpretation of STING's role in ISG regulation and tumor biology.

4. Regarding the new ChIP-seq data for CBF- β , the results appear highly unusual. Across seven chromosomes, only a few peaks were detected, which is atypical. A quick literature search indicates that ChIP-seq datasets for RUNX proteins and CBF- β typically yield thousands to tens of thousands of peaks. For a transcriptional regulator, such sparse peak detection often suggests a failure in achieving specific or efficient ChIP enrichment.

Unfortunately, it is not currently possible to assess the quality of this dataset, as the ChIP-seq data (GSE270776) remain private, even when accessed with the provided token: upszseuqvbyddsp. In contrast, the CRISPR screening dataset (GSE270775; token: anwvmykobbstcl) is accessible.

To better evaluate the quality of the CBF- β ChIP-seq data, could the authors clarify the total number of peaks detected using an enrichment threshold of 5 and FDR < 0.01? This information would be critical for assessing the reliability and interpretability of the dataset.

5. Regarding Major Concern 1 (as raised in the rebuttal), this issue remains unaddressed, possibly due to a miscommunication. While it is acceptable that type I IFN does not fully explain the observed synthetic lethality—as the authors have acknowledged—this was not the central point of the previous review.

The key concern is that the authors claim CBF- β knockout is synthetically lethal in VHL-deficient cells and CBF- β directly regulates STING. Despite the efforts to establish both types of regulation, the authors have not provided any data connecting the synthetic lethality to type I IFN regulation. This gap is particularly significant given that both synthetic lethality and type I IFN signaling are emphasized in the title and represent two major themes of the study.

As a transcriptional regulator, CBF- β influences a wide array of cellular pathways. Without genetic evidence directly linking type I IFN signaling to the synthetic lethality phenotype, the observed regulation may be incidental. To address this, the authors could perform cell death and in vivo tumor assays—such as those shown in Fig. 2—using STING and IRF3 wild-type and knockout VHL-null ccRCC cells, which should be straightforward to implement.

In its current form, the manuscript presents two distinct regulatory themes—synthetic lethality and type I IFN signaling regulated by CBF- β —without establishing a mechanistic connection between them.

6. Furthermore, in rebuttal point 1, the authors propose: “It is therefore plausible that VHL-null cells may have a basal degree of mitochondrial instability, rendering them susceptible to mitochondrial insults and activation of the cGAS-STING pathway (in the context of CBF β knockout).” However, the more pertinent and unresolved question is how CBF β knockout activates cGAS, a cytosolic sensor of double-stranded DNA.

The authors state that CBF β knockout does not lead to mitochondrial dysfunction or genomic DNA damage (new Extended Data Fig. 6a–h). In the absence of evidence for DNA leakage or damage, the mechanism underlying cGAS activation remains unclear. This gap is critical, as it directly affects the interpretation of how CBF β loss engages the cGAS-STING pathway in VHL-null cells and undermines the proposed model.

7. Regarding rebuttal point 3, the authors present data in Fig. 6e (right panel) showing that cGAS sgRNA reduces STING mRNA induction. However, this appears to be a red herring. The appropriate approach to assess whether cGAS regulates STING activation is to perform a cGAS knockout and examine phosphorylated STING or TBK1, as cGAS primarily regulates STING post-transcriptionally, consistent with the signal transduction model illustrated in Fig. 6a and Fig. 8.

As currently presented, the manuscript begins with a mechanistic framework in Fig. 6a, yet diverges from it by investigating transcriptional regulation of STING mRNA by cGAS. If the authors intend to demonstrate that cGAS regulates STING transcriptionally, this contradicts the premise of Fig. 6a and reflects a conceptual inconsistency. Conversely, if they conclude that cGAS does not regulate STING mRNA, this conflicts with the data shown in Fig. 6e (right panel), which indicates a statistically significant reduction.

In either case, the data concerning cGAS are problematic, and the mechanism of STING activation remains insufficiently addressed. Moreover, the authors’ interpretation does not resolve the discrepancy between their conclusions and the presented data. In the text (lines 292–293), they state: “cGAS knockout did not affect CBF- β knockout-induced STING expression (Fig. 6e and Extended Data Fig. 6i).” Yet Fig. 6e (right panel) clearly shows a statistically significant difference ($p=0.012$), which contradicts this claim.

8. In this revision, the authors attempt to conclude that IFN- β (type I IFN) is not involved and propose a model for the direct regulation of ISGs by IRF3 (Fig. 5). However, the data presented—such as those concerning STAT1/STAT2—are suggestive but not definitive. Direct evidence would require demonstrating that IFN β gRNA has no effect, which has not been shown. JAK kinases have been shown to activate other pathways, such as PI3K-AKT and RAS-MAPK Pathway.

Minor Points:

1. In Figure 3, the objective and the title is misaligned. The primary goal of Figure 3, as stated in the text (line 163), is to demonstrate that both CBF- β and RUNX proteins contribute to the synthetic lethal interaction with VHL. However, the figure title seems to focus on how RUNXs are regulated by CBF- β at the translational level. This emphasis appears to diverge from the main theme of the manuscript. As the authors themselves noted, the post-translational regulation of RUNXs by CBF- β has already been addressed in a previous publication (Malik et al., 2020, Nature Communications). It is recommended that the author stick to their main topic.

2. Clarification is needed for “Immune Cell-Specific IRF7”: The phrase “immune cell-specific IRF7” is ambiguous. Are the authors implying that IRF7 expression is restricted to immune cells? This interpretation appears inconsistent with their own data in Extended Data Figure 5e, which shows IRF7 expression in cancer cells. Further clarification is warranted to resolve this confusion.

Reviewer #3

(Remarks to the Author)

Since the author has provided clarifications and additional experiments in response to the reviewers’ concerns, the manuscript is now deemed acceptable for publication upon re-evaluate.

Reviewer #4

(Remarks to the Author)

The authors have addressed the majority of my concerns.

However, Figure 2h should be shown as the tumor volume $\text{tumor volume (mm}^3\text{)} = (\text{length} \times \text{width}^2)/2$, not tumor area.

Reviewer #5

(Remarks to the Author)

Berlin et al describe a series of genetic screens ostensibly to discover genes that are essential in the context of VHL loss,

the signal genomic lesion in kidney cancer (clear cell renal cell carcinoma, ccRCC). The authors perform genome-scale CRISPR knockout screens in two ccRCC cell lines, 786O and RCC4, which carry the signature homozygous VHL loss, and again in those same cells with exogenously expressed VHL.

It is not clear how well-controlled the CRISPR screens are. The authors use the TKOv3 one-component library. Some quality control figures are present in Supp Fig 1, but the relative paucity of hits (genes with Bayes Factor > 0 or some higher threshold) shown in Fig 1C suggests that quality was not optimal. It would be useful to have a precision-recall curve, using common reference sets of essential and nonessential genes (e.g. for essentials, Hart et al, G3, 2017 – the Hart et al, Cell 2015 set is not optimal – and for nonessentials, Hart et al MSB 2014). Other data, such as cell-line specific nonexpressed genes, may be substituted for nonessentials without loss of information. Comparison of the PR curves of these screens to those publicly available ccRCC genome-scale cell line screens in DepMap would provide further evidence of screen quality. This information is important, as it gives some insight into the false negative rate of a cell line screen, which is critical in evaluating whether a hit found in one screen but not another reflects underlying biology or just a missed call in the other screen.

We thank this reviewer for their assessment of our work. We did not include all the control figures, but please see below the precision-recall (PR) curves using the reference sets for non-essential and essential genes as suggested. These show the PR values improving over the duration of the screens relative to the sequenced plasmid, as described in Hart and Moffat, BMC Bioinformatics, 2016 (PMID: 27083490).

I am satisfied with this analysis for the PR curves.

I am wondering what the authors are plotting in Fig. 1C. Both axis say “Bayes Factor” and the cell line screen. It appears they are actually plotting $BF(VHL^{-/-}) - (VHL^{+/+})$ to find VHL SL interaction that are common to both cell lines. This will need to be clarified. If they are indeed plotting BF then I agree, the SL interactions they identified are relatively weak since the BF between VHL^{-/-} and VHL^{+/+} are relatively low. Since the remainder of the manuscript is validating CFBF then I think the screen holds for the purpose of this manuscript.

We have updated the Venn diagrams displayed in Extended Data Fig. 1c, to include the core essential gene set described in Hart et al., G3, 2017 (PMID: 28655737). Again, these show robust discrimination of essential genes.

I am satisfied with the Venn diagrams

We accept that there will be false negatives from our approach, but our aim was to identify potential synthetic lethal interactors of VHL which may be pertinent to ccRCC therapy, rather than to provide a comprehensive atlas of VHL-associated synthetic lethality.

I agree with the authors on this point, they are not looking to catalog all the SL interactions between with VHL-LOF, they are simply trying find some that are present in both cell lines which they have accomplished.

To identify genes providing proliferative advantage (e.g. KEAP1 in this case), a better approach is described in Lenoir et al, Nat Comms 2021. This is far better than Bagel scores, although the outlier nature of KEAP1 knockout is a strong signal here.

We have analysed our data with both DrugZ and MAGeCK to obtain enrichment scores for genes that improve survival. Please see the results below (also presented in Supplementary Table 1), which show that KEAP1 sgRNAs are by far the most enriched, regardless of the analysis workflow employed.

I am satisfied with the DrugZ and MAGeCK analysis.

The cherry-picking of target genes (Fig 1d) is useful for narrowing the reader's attention to the genes of interest but the display is not convincing. It is standard to calculate and display fold change of normalized end point cells relative to some early point (plasmid or T0), rather than read counts, alongside some negative control (again, nonessentials are useful here) whose distribution is centered around zero.

We elected to display the trajectory of normalised sgRNA read counts as this includes both the desired fold change information, as well as a central timepoint to provide additional temporal information about the depletion/enrichment of sgRNAs over the course of the screens. We have added the sgRNAs targeting EGFP as negative controls to Fig. 1d. As expected, these show minimal change, with only mild enrichment in RCC4 cells as a result of the dropout of essential genes from the pool.

I agree with reviewer 1 that displaying the normalized sgRNA read count trajectory is not typical. However, Fig. 1d is essentially another way to display the screen data and is not critical for the remaining scope of the manuscript.

For comparing differential essentials between cell lines, the approach is flawed. The difference in proliferation rates between the VHL⁺ and VHL⁻ cells will confound a competition assay for single gene effects. A better approach would be to use an algorithm such as DrugZ (Colic et al, Genome Med 2019; see, e.g., Karasu et al Nat Biotech 2024 for a recent example of its application) to compare endpoint readcounts of screens in VHL⁺ and VHL⁻ cells.

We had already used MAGeCK but chose to display the finding with BAGEL2. We have now analysed the data with DrugZ

and neither of these additional methods alter our findings: as displayed below, CBFb remains a top hit within all of these analyses. Full results are given in Supplementary Table 1.

I am satisfied with the DrugZ analysis

The competition assays (Fig 1. f,g) are normalised to account for differential proliferation rates, and mCherry/GFP-expressing clonal cell lines were selected to minimise proliferation differences. Furthermore, the fact that guides targeting CBFb and KEAP1 produce opposite effects in these assays further reinforces our conclusions in this initial validation experiment.

I am also not convinced by the pooled competition assay result. Reviewer 1 is correct, the differential growth rates of VHL-/- vs. VHL+ will confound the assay and is very hard to control for. Typically, these types of competition assays are done with one cell line comparing control sgRNA to another sgRNA. Colony formation assays are a better way to validate the SL interaction when analyzing two isogenic cell lines with different proliferation rates and should be considered.

My motivation for stressing controls so firmly is that the main gene chosen for follow up, CBFb, is a very weak hit in Fig 1d. More importantly, it is not a hit in DepMap. In fact CBFb knockout shows strong phenotype in leukemia, lymphoma, and myeloma cells (Chronos score < -0.5), but is never essential in any of the >25 kidney cancer cell lines screened by DepMap. The in vivo data is compelling, but is totally inconsistent with the in vitro data that were ostensibly used to nominate CBFb for validation.

We appreciate that this reviewer found the in vivo data compelling, but we respectfully disagree with their assertion regarding the screens. We set out to take an unbiased strategy to look for experimentally find synthetic lethal interactors, which is exactly why we used the dropout CRISPR approach. As would be important for any screening approach, we go on to validate our findings. The screening data is entirely consistent with the subsequent validation. Several genes and pathways known to be important in targeting renal cell cancer are not hits in DepMap (e.g. TBK1, ROCK1, FTO, SCARB1), and DepMap is not a substitute for undertaking a screen based on VHL deficiency or proficiency. As discussed previously (PMID: 34244212), differences between studies may arise due to the use of different cell culture media. Other possible reasons include the employment of different ccRCC cell lines and CRISPR/RNAi libraries in screening experiments.

While CBFb is a weak hit in the screen, I agree with the authors that lack of signal in DepMap does not invalidate their results. One possible explanation is due to the artificial nature of adding VHL back to cancer cell lines. While targeting CBFb in ccRCC may not be a feasible approach in the clinic, the authors have still found some potentially interesting biology to explore based on their screen results.

The authors should either perform a more realistic analysis of their screen data, or find another way to get to the CBFb hypothesis in ccRCC.

We disagree with this statement, and have performed a thorough analysis of the screen data. Experimental validation is always important following a screen: we have extensively demonstrated the lethal effect of CBF- β loss in ccRCC in vitro and in vivo, which is further supported by the overexpression of CBFb in ccRCC tumours.

I am satisfied with the reviewer response.

Version 2:

Reviewer comments:

Reviewer #6

(Remarks to the Author)

I find the manuscript to be technically well-executed, detailed, and to have identified a novel dependency in RCC that could be therapeutically relevant.

There are a couple issues that need to be addressed prior to publication:

Figure 8 does not adequately represent the findings of the manuscript as only the transcriptional function of CBFb is included. The figure should be modified to show both the transcriptional and translational roles (upregulation of RUNX1 in particular) of CBFb.

CBFb ChIPSeq has been performed and many peaks were identified (see for example <https://doi.org/10.1371/journal.pgen.1005884>) as Reviewer 2 indicated, so the authors' response to the critique about the lack of peaks in their CBFb ChIPSeq data is not compelling. Either the ChIPSeq should be repeated to achieve higher quality data or ChIP-qPCR should be done for the STING locus to confirm the presence of CBFb.

Version 3:

Reviewer comments:

Reviewer #6

(Remarks to the Author)

The author has addressed my concerns. The manuscript is acceptable for publication.

Response to Reviewers' Comments

We thank the reviewers for their very helpful and constructive feedback. Please find below our responses to the points made.

Reviewer #1 (Remarks to the Author):

Bertlin et al describe a series of genetic screens ostensibly to discover genes that are essential in the context of VHL loss, the signal genomic lesion in kidney cancer (clear cell renal cell carcinoma, ccRCC). The authors perform genome-scale CRISPR knockout screens in two ccRCC cell lines, 786O and RCC4, which carry the signature homozygous VHL loss, and again in those same cells with exogenously expressed VHL.

It is not clear how well-controlled the CRISPR screens are. The authors use the TKOv3 one-component library. Some quality control figures are present in Supp Fig 1, but the relative paucity of hits (genes with Bayes Factor > 0 or some higher threshold) shown in Fig 1C suggests that quality was not optimal. It would be useful to have a precision-recall curve, using common reference sets of essential and nonessential genes (e.g. for essentials, Hart et al, G3, 2017 – the Hart et al, Cell 2015 set is not optimal – and for nonessentials, Hart et al MSB 2014). Other data, such as cell-line specific nonexpressed genes, may be substituted for nonessentials without loss of information. Comparison of the PR curves of these screens to those publicly available ccRCC genome-scale cell line screens in DepMap would provide further evidence of screen quality. This information is important, as it gives some insight into the false negative rate of a cell line screen, which is critical in evaluating whether a hit found in one screen but not another reflects underlying biology or just a missed call in the other screen.

We thank this reviewer for their assessment of our work. We did not include all the control figures, but please see below the precision-recall (PR) curves using the reference sets for non-essential and essential genes as suggested. These show the PR values improving over the duration of the screens relative to the sequenced plasmid, as described in Hart and Moffat, BMC Bioinformatics, 2016 (PMID: 27083490).

We have updated the Venn diagrams displayed in **Extended Data Fig. 1c**, to include the core essential gene set described in Hart et al., G3, 2017 (PMID: 28655737). Again, these show robust discrimination of essential genes.

We accept that there will be false negatives from our approach, but our aim was to identify potential synthetic lethal interactors of *VHL* which may be pertinent to ccRCC therapy, rather than to provide a comprehensive atlas of *VHL*-associated synthetic lethality.

PR curves for CRISPR screens in 786O and RCC4 backgrounds at early, intermediate and late timepoints, relative to the plasmid.

To identify genes providing proliferative advantage (e.g. KEAP1 in this case), a better approach is described in Lenoir et al, Nat Comms 2021. This is far better than Bagel scores, although the outlier nature of KEAP1 knockout is a strong signal here.

We have analysed our data with both DrugZ and MAGeCK to obtain enrichment scores for genes that improve survival. Please see the results below (also presented in **Supplementary Table 1**), which show that *KEAP1* sgRNAs are by far the most enriched, regardless of the analysis workflow employed.

Pairwise analyses by DrugZ (left) and MAGeCK (right) of the late timepoints of CRISPR/Cas9 screens, showing gene knockouts providing a proliferative advantage in the VHL-null background.

The cherry-picking of target genes (Fig 1d) is useful for narrowing the reader's attention to the genes of interest but the display is not convincing. It is standard to calculate and display fold change of normalized end point cells relative to some early point (plasmid or T0), rather than read counts, alongside some negative control (again, nonessentials are useful here) whose distribution is centered around zero.

We elected to display the trajectory of normalised sgRNA read counts as this includes both the desired fold change information, as well as a central timepoint to provide additional temporal information about the depletion/enrichment of sgRNAs over the course of the screens. We have added the sgRNAs targeting *EGFP* as negative controls to Fig. 1d. As expected, these show minimal change, with only mild enrichment in RCC4 cells as a result of the dropout of essential genes from the pool.

For comparing differential essentials between cell lines, the approach is flawed. The difference in proliferation rates between the VHL+ and VHL- cells will confound a competition assay for single gene effects. A better approach would be to use an algorithm such as DrugZ (Colic et al, Genome Med 2019; see, e.g., Karasu et al Nat Biotech 2024 for a recent example of its application) to compare endpoint readcounts of screens in VHL+ and VHL- cells.

We had already used MAGeCK but chose to display the finding with BAGEL2. We have now analysed the data with DrugZ and neither of these additional methods alter our findings: as displayed below, *CBFB* remains a top hit within all of these analyses. Full results are given in **Supplementary Table 1**.

The competition assays (Fig 1. f,g) are normalised to account for differential proliferation rates, and mCherry/GFP-expressing clonal cell lines were selected to minimise proliferation differences. Furthermore, the fact that guides targeting *CBFB* and *KEAP1* produce opposite effects in these assays further reinforces our conclusions in this initial validation experiment.

Pairwise analyses by DrugZ (left) and MAGeCK (right) of the late timepoints of CRISPR/Cas9 screens, showing genes which are synthetic lethal with VHL.

My motivation for stressing controls so firmly is that the main gene chosen for follow up, *CBFB*, is a very weak hit in Fig 1d. More importantly, it is not a hit in DepMap. In fact *CBFB* knockout shows strong phenotype in leukemia, lymphoma, and myeloma cells (Chronos score < -0.5), but is never essential in any of the >25 kidney cancer cell lines screened by DepMap. The in vivo data is compelling, but is totally inconsistent with the in vitro data that were ostensibly used to nominate *CBFB* for validation.

We appreciate that this reviewer found the in vivo data compelling, but we respectfully disagree with their assertion regarding the screens. We set out to take an unbiased strategy to look for experimentally find synthetic lethal interactors, which is exactly why we used the dropout CRISPR

approach. As would be important for any screening approach, we go on to validate our findings. The screening data is entirely consistent with the subsequent validation. Several genes and pathways known to be important in targeting renal cell cancer are not hits in DepMap (e.g. TBK1, ROCK1, FTO, SCARB1), and DepMap is not a substitute for undertaking a screen based on VHL deficiency or proficiency. As discussed previously (PMID: 34244212), differences between studies may arise due to the use of different cell culture media. Other possible reasons include the employment of different ccRCC cell lines and CRISPR/RNAi libraries in screening experiments.

The authors should either perform a more realistic analysis of their screen data, or find another way to get to the CBFb hypothesis in ccRCC.

We disagree with this statement, and have performed a thorough analysis of the screen data. Experimental validation is always important following a screen: we have extensively demonstrated the lethal effect of CBF- β loss in ccRCC *in vitro* and *in vivo*, which is further supported by the overexpression of *CBFB* in ccRCC tumours.

Reviewer #2 (Remarks to the Author):

In this study, the authors performed synthetic lethal screens and identified CBFb deletion as synthetic lethal to VHL loss. Mechanistically, they propose that CBFb negatively regulates STING. Overall, the finding that CBFb and VHL are synthetic lethal is potentially interesting. However, the mechanism underlying the role of CBFb in this synthetic lethality remains unclear, and some of the data do not support the model and appear to be conflicting. Additionally, the manuscript's writing needs improvement, as some data contradict the conclusions presented in the figures. The primary concerns are listed below. If the authors can sufficiently address these concerns, I am willing to review a revised version of the manuscript.

We appreciate this reviewer's interest in the work and our comments to their concerns are outlined below.

Concerns:

1. A major issue about the study is that the results have not provided reasonable mechanistic explanation for the synthetic lethality of CBFb and VHL loss, the major topic of this study. Type I interferon activation induces a plethora of phenotypes in cells, including senescence, apoptosis, cell cycle arrest, etc. Why does CBFb deletion-induced type I IFN activation selectively kill VHL-null ccRCC? Without the mechanism, the main conclusion is not well supported.

We accept that the induction of a Type I IFN response does not fully explain why CBF- β depletion is synthetically lethal with VHL, and we already noted this in the manuscript. However, we disagree that our main conclusions are not supported. We show evidence *in vitro* and *in vivo* that CBF- β loss results in decreased tumour growth in a VHL-dependent manner, and our data also demonstrates that CBF- β is a negative regulator of STING-mediated ISG induction. We did not conclude that Type I IFN activation was solely sufficient to drive cell death in the *VHL*-null cells, and have clarified the text in this regard, and well as the findings of **Extended Data Fig. 5i-k**.

We now also provide further mechanistic explanations as to why *VHL*-null cells are more sensitive to *CBFB* knockout, and have investigated whether DNA damage and mitochondrial function is affected in a *VHL*-dependent manner (**Extended Data Fig. 6d-f**). Interestingly, we observe that mitochondrial DNA leakage is higher in *VHL*-null cells following *CBFB* knockout (**Extended Data Fig. 6d**). This is in-keeping with a recent report that demonstrates higher mtDNA leak following *VHL* knockout in mouse colon

adenocarcinoma cells (PMID: 39050705). It is therefore plausible that *VHL*-null cells may have a basal degree of mitochondrial instability, rendering them susceptible to mitochondrial insults and activation of the cGAS-STING pathway. We have modified the text to explain this point.

2. The increase in STING protein alone is not sufficient to activate the type I interferon pathway. Therefore, *CBFB* deletion must activate an upstream regulator of STING. The authors should employ a candidate approach to knock out potential upstream activators and elucidate the mechanism of type I IFN activation. The strategy outlined in the Cell paper, "Inhibiting DNA Methylation Causes an Interferon Response in Cancer via dsRNA Including Endogenous Retroviruses," may be helpful to the authors (ScienceDirect, Volume 162, Issue 5, 27 August 2015, Pages 974-986).

The reviewer already notes that we have shown that cGAS is required (see point 3) and therefore we agree that upstream activation of STING occurs. We also show that transfection of dsDNA following *CBFB* loss dramatically increases the type I IFN response (Fig. 6d). However, our main point in the manuscript is that CBF- β regulates the level of STING, which delineates the extent of the type I IFN response. This is entirely consistent with the original characterisation of STING activation by cGAMP (PMID: 21947006). We have revised the text accordingly, and highlight that the discovery of STING as an inducer of interferon signalling was made by modulating levels of STING protein alone (PMID: 18724357). We have also amended the results of **Extended Data Fig. 7a** to clarify that, in this experimental context, *CBFB* deletion must also activate an upstream regulator of STING, as the reviewer suggests.

We thank the reviewer for highlighting the DNMT inhibitor paper, which we were aware of. We considered whether ERVs may be relevant to the phenotype, and had already examined our RNA-Seq data to see if there was any upregulation, but no clear changes were apparent (**Supplementary Table 4**). We have now also performed qPCR for ccRCC-associated ERVs, but again did not observe any changes upon *CBFB* knockout (**Extended Data Fig. 6h**).

3. The data suggest that cGAS, an upstream activator of STING, is required for type I IFN activation upon *CBFB* depletion (Figure 6e). As cGAS is a double-stranded DNA (dsDNA) sensor, the authors should conduct experiments to identify the source of dsDNA.

As discussed above, we have expanded our extensive search for the source of the dsDNA, including examining whether CBF- β loss results in DNA damage, mitochondrial DNA leakage, or ERV expression (**Extended Data Fig. 6**).

4. The whole manuscript is about ccRCC, and the GFP-Vif section in Figure 7 is irrelevant to the study. While it is acceptable if the authors aim to use GFP-Vif to degrade *CBFB*, this approach does not add significant value beyond the use of sgRNA or shRNA for *CBFB*. It remains unclear why the authors conducted an HIV-related study and how they concluded that the *CBFB* amount is lower than the basal conditions (as shown in the bottom of Figure 7).

We used Vif not to degrade CBF- β but to sequester CBF- β , as it forms part of an E3 ligase complex with Vif. Therefore, it is a complementary approach to demonstrate that sequestering CBF- β will also result in increased STING levels, and that this activity has broad potential implications beyond the kidney cancer field. We would therefore prefer to include this work within the manuscript.

5. It is puzzling that the induction of ISGs does not require STAT1, STAT2, and IRF9 (Extended Data Fig. 4a). On page 7, first paragraph, it states, "To our surprise, however, knockout of STAT1, STAT2, or IRF9 individually, or in combination, did not prevent IFIT1 induction (Fig. 5b and Extended Data Fig. 4a-c)." One possibility is that the gRNAs for STAT1, STAT2, and IRF9 generated a polyclonal pool, with some

cells having reduced STAT1, STAT2, or IRF9 levels, and others remaining wild type. This polyclonal nature could result in the negative outcome, particularly in the context of the autocrine and paracrine effects of IFN- β . The authors should select at least two single clones for STAT1, STAT2, and IRF9 knockout and repeat the experiment in Extended Data Fig. 4a.

We were also surprised that STAT1 and STAT2 were not involved, but **Fig. 5b** already demonstrates that the depletion of STAT1 and STAT2 ablates the Type I IFN response, as the cells were completely unresponsive to IFN- β treatment. Therefore, we see no reason to repeat the experiments as suggested, as we already show that our STAT1/STAT2 double knockout is highly efficient.

6. For Extended Data Fig. 4, the title is “ISG transcription following CBF β deletion is mediated by the STING-TBK1-IRF3 axis.” However, the data appear to suggest otherwise. Please clarify this discrepancy.

We apologise if we are missing something here, but we see no discrepancy here, as our findings show that ISG induction is dependent on a STING-TBK1-IRF3 axis.

7. In Figure 4e, CBF β N104A rescues the effect of IFIT1 similarly to WT, suggesting that binding to RUNX proteins is not necessary for the repression of type I interferon. This result contradicts the model proposing that the CBF β -RUNX complex suppresses STING. It has been established that CBF β enhances the chromatin binding of RUNX proteins. Therefore, reconciling this result with the proposed model requires a logical explanation.

We appreciate the uncertainty around this point, and have considered a number of plausible explanations. First, in **Fig. 3**, we confirmed the findings of Malik et al. (2019) (PMID: 31061501) that CBF- β also regulates the translation of RUNX proteins: it may be that, in this context, CBF- β -RUNX dimerisation is dispensable for the regulation of STING, which can be achieved by RUNX alone, and that the primary effect of CBF β knockout is rather to modulate total RUNX levels. Alternatively, there may still be a small amount of residual binding between the CBF- β -N104A mutant and RUNX proteins, which we have been unable to detect.

To explore these possibilities further, we have overexpressed RUNX1 and RUNX2 and identified that RUNX1 in particular is capable of STING repression even in the absence of CBF- β , but that its effects are greatest in the presence of CBF- β (**Fig. 7c,d**). As we explain in the revised text, this suggests that RUNX1 (and to a lesser extent RUNX2) is indeed the primary effector of STING repression, CBF- β dimerisation is likely to be required for maximal efficiency. This would also be consistent with the small difference between WT and N104A CBF- β in the figure highlighted by the reviewer (**Fig. 4e**).

8. In Figure 6f, the classical ChIP analysis has several issues. Firstly, it was heavily normalized. The authors should present the percentage of input on the Y-axis. Secondly, ChIP-seq or CUT&RUN analyses of CBF β or RUNX2 should be performed to unbiasedly detect the binding of CBF β and RUNX2 genome-wide.

We have modified our analysis of ChIP-qPCR experiments to display the % of input, as suggested (**Extended Data Fig. 5g**). We have also undertaken ChIP-Seq of CBF- β , alongside using overexpressed CBF- β to control for specific binding with the antibody. The findings of this CBF- β chromatin occupancy were quite remarkable, as *STING* was in fact the dominant peak across the whole genome within our ChIP-Seq analysis (**Fig. 7a,b**). However, CBF- β did not associate with other ISGs (**Fig. 7a**). These findings strongly support the involvement of CBF- β in regulating the *STING* locus.

9. The claim that RUNX1 directly binds to the promoter region of STING is questionable because the ChIP-seq data were obtained from a public dataset involving RUNX1 in K562 cells (chronic myeloid leukemia). Transcriptional and epigenetic features are cell-line specific. Thus, binding of RUNX1 in K562 cells does not necessarily indicate binding in ccRCC cell lines. The authors should perform ChIP-seq analysis of RUNX1 in ccRCC cell lines to substantiate this claim.

We thank the reviewer for this comment, and we have now removed the RUNX1 ChIP-seq from K562, and instead focused on the CBF- β ChIP-seq from the 786O, as detailed above. As our data have already examined the involvement of RUNX1 in conjunction with CBF- β in ccRCC, additional RUNX1 ChIP-seq will be of limited value and therefore we have not pursued this further.

10. The authors claim that IRF3 directly induces ISGs, which contradicts existing literature. Therefore, robust data are needed to support this claim. IRF3 ChIP-seq data is required to substantiate the assertion. The authors have only performed an IRF3 knockout and demonstrated that IRF3 is necessary, which this piece of data does not indicate IRF3 directly induce ISGs.

We disagree with the reviewer's point, as there is a substantial body of evidence that IRF3 homodimers are capable of directly stimulating ISG transcription independently of interferons. For example, cytomegalovirus infection can induce ISG transcription through IRF3 even in the presence of protein synthesis inhibitors or IFN- β -neutralising antibodies (PMID: 9391139, PMID: 10950979, PMID: 16379004, PMID: 30871003). IRF3 binding to the interferon-stimulated response elements (ISREs) of canonical target genes has also been demonstrated previously (PMID: 31799619). We have clarified the involvement of IRF3 induction of ISGs within the text. In addition, we have performed ChIP-qPCR of IRF3 at the ISREs of *IFIT1*, *IFIT2*, and *OASL*, which demonstrates increased binding upon CBF- β knockout (**Extended Data Fig. 5g**).

Reviewer #3 (Remarks to the Author):

This study offers a novel and interesting insight into the mechanism of "synthetic lethality" in VHL-null kidney cancers. In this manuscript, the authors utilized a genome-wide CRISPR knockout screen to identify the synthetic lethality interactors of von Hippel-Lindau (VHL) in clear cell renal cell carcinoma (ccRCC). They found that the loss of CBF- β suppresses the growth of VHL-null tumors both in vitro and in vivo. Mechanistically, the authors demonstrated that the CBF- β /RUNX complex binds to the STING promoter, inhibiting STING transcription. The loss of CBF- β leads to cell-intrinsic STING activation, resulting in the upregulation of type I interferon and the expression of interferon-stimulated genes (ISGs), which ultimately arrests the cancer cell cycle. This research demonstrated innovation and provided a potential therapeutic target for ccRCC. However, more robust and convincing evidence is required to support the hypothesis.

We thank the reviewer for their appreciation of the work, recognition of the innovative approach, and their helpful suggestions.

Major comments :

1. Previous studies (Refs. 24-34 mentioned in the manuscript) have explored various 'synthetic lethality' interactors of VHL. So, what is the advantage of this study or the new target of CBF-b. Whether other synthetic lethality interactors of VHL (eg. ROCK1, J M Thompson, et. al, Oncogene, 2017) also have similar functions in the ccRCC case? What is the specificity of CBF-b as the 'synthetic lethality' interactors of VHL in the context of ccRCC?

The requirement for VHL loss in driving ccRCC presents an excellent candidate for synthetic lethal interactions. However, previous studies have often used focussed siRNA libraries or single ccRCC cell lines to interrogate synthetic lethal interactions. Hence, our approach outlined here, uses different ccRCC lines and with genome wide libraries. CBF- β , being a top candidate in the screen, provides a novel approach to target VHL loss and the dependency in ccRCC. Whether other, previously identified synthetic lethal interactions should function within the same CBF- β pathway is an area that we are currently pursuing but it beyond the current scope this manuscript. Interestingly, ROCK1 has been reported to modulate IRF3 via phosphorylation (PMID: 34533996), so it is certainly plausible that it may have effects within the IFN pathway. However, the notion of synthetic lethality does not mean that all genes need to function within the same pathway.

To further address why *VHL* loss confers susceptibility to *CBFB* loss, we now present data which show that VHL appears to protect cells from mitochondrial damage, as the leakage of mitochondrial DNA is specifically reduced in VHL expressing cells following CBF- β loss (**Extended Data Fig. 6**), consistent with a recent study in a mouse colon adenocarcinoma model (PMID: 39050705). It is therefore possible that this relative abundance of mitochondrial dsDNA within the cytosol makes *VHL*-null cells more susceptible to CBF- β depletion.

2. Initially, the author aimed to find the potential therapeutic targets for ccRCC and screened out CBF-b as a 'synthetic lethality' interactor of VHL. Can the author target CBF-b/RUNX-STING-IRF3-IFN axis for the treatment of ccRCC in a mouse model?

We agree that this would be interesting but there are currently no appropriate specific inhibitors of CBF- β . We also do not have a GEM-ccRCC model to test this, as an immune competent mouse would be important. Establishing such models is a longer-term goal, beyond the scope of this work. However, recent work from the Carugo and Genovese groups (PMID: 37365326) using a GEM-RCC model identify downstream signalling from the IFN receptor as an important suppressor of RCC tumour growth, which is entirely consistent with our conclusions. In addition, the Blyth group have shown in different GEM-RCC models that RUNX1 loss ablates tumour growth (PMID: 32156779). Whilst we did reference these studies, we have edited the text accordingly to explain their relevance to our findings.

3. In the manuscript, the author focuses solely on tumor cells. If the loss of CBF-b activates the STING signaling pathway in ccRCC, how does this affect other immune cells in the tumor microenvironment (TME)? Additionally, what impact could this have on T cells, natural killer (NK) cells, or other immune cell populations in systemic conditions?

This is an interesting question but would require a GEM-ccRCC model and such work is clearly beyond the scope of this manuscript. We have edited the text to emphasise the important nature of this reviewer's point and to explain that it will be the subject of ongoing work.

4. The author claimed that the loss of CBF-b in VHL-null 786O cells activates STING-IFN-ISG signaling. If the 786O cells produce and secrete type I interferon (IFN), why does the secreted IFN not affect the 786O+VHL cells in the co-culture experiment (Fig. 1e)?

We apologise for the confusion here. CBF- β loss leads to upregulation of ISGs but we did not detect Type I IFN secretion in the 786O cells. IFN- β was only activated following transfection of exogenous dsDNA. This was discussed in the text, and hence we referred to the induction of ISGs as 'cell-intrinsic' but we have clarified this point.

Could it be that STING activation triggers type I IFN-independent signaling pathways, such as autophagy and autophagy-mediated cell death?

We had already tested whether Bafilomycin A1 treatment, which would alter autophagic flux, provided a relative protection to 786O cells following CBF- β depletion, but observed no difference. We have included this data as **Extended Data Fig. 3e**.

5. In Fig 1a, VHL was reconstituted in the 786O and RCC4 cell lines. Why do the Western blot bands differ in these two cell lines?

We often see different translational processing of VHL between cell lines. The functions of these isoforms are largely overlapping (PMID: 9751722), as indeed we demonstrate in **Fig. 1a** with respect to HIF regulation. We have updated the text to explain this.

6. In Fig. 5g, why is the STING protein absent in the 786O Cas9 (sgCtrl) cells?

STING is still present but only observed at higher exposures. We have included a higher exposure film for comparison (**Fig. 5g**).

7. In Ex.Fig.6b and 6c, the knockout of CBF-b in RCC4 cells increases the mRNA level of cGAS and STING. However, in RCC10 cells, the knockout of CBF-b appears to have no effect on the regulation of either cGAS or STING levels. This suggests that the idea that CBF-b/RUNX binds to the STING promoter region to repress STING transcription may not be a universal model.

Both the RCC4 and RCC10 cells show the same trend, with P values of 0.019 compared to 0.066, so we do not see a particular concern here. However, we have not claimed that our findings would be universal. In fact, a universal action would be surprising: indeed HIF-2 α inhibitors, which are now in clinical use, did not show universal activity in ccRCC pre-clinical models (PMID: 27595393).

8. Why did the author use the ChIP-Sequencing data from the K562 cell line instead of the ccRCC?

We chose to interrogate publicly available DNA-binding data for CBF- β , which happens to have been performed in K562, before validating this ourselves in RCC cells. However, as this is a point raised by a number of reviewers, we have now run our own ChIP-Seq to demonstrate that CBF- β does indeed strongly bind the *STING* locus in 786O cells (**Fig. 7a**).

9. In Fig. 6f, ChIP analysis using the CBF-b antibody showed that the levels of STING primers 1 and 3 were reduced in CBF-b KO cells. However, when using the RUNX2 antibody, only primer 1 exhibited a reduction, while primer 3 did not. Why? Does this suggest that primer 1 is more significant? Additionally, why is the sample size for primers 1 and 2 listed as n=5, while for primers 3 and 4, it is n=3?

We have now included CBF- β ChIP-seq from the 786O cells to measure CBF- β chromatin occupancy in an unbiased way. We have therefore removed the RUNX ChIP-PCR data, as this was an indirect measure of how CBF- β might bind.

Minor point :

1. Some of the western blot bands are too faint. For example, in Fig 4e, the band of IRF7 is barely visible, especially for the sg control.

We have amended **Fig. 5g**, as discussed above in response to point 6. Regarding the IRF7 immunoblot (now **Extended Data Fig. 5e**), there is essentially no IRF7 in these cells, which was expected as IRF7 is only expressed in immune cells. We have edited the text to explain this. We would be happy to present higher exposures of other films if the reviewer wishes.

Reviewer #4 (Remarks to the Author):

The manuscript by Bertlin et al. identifies the CBF-B transcriptional regulator as a synthetic lethal partner of VHL loss in kidney cancer cells. In this study, performed a pooled genome wide CRISPR/Cas9 screen to identify synthetic lethal partners of VHL loss in 2 kidney cancer cell lines. They demonstrate that loss of CBF-B selectively reduces the growth and survival of VHL deficient kidney cancer cell lines. Importantly, genetic inhibition of CBF-B reduces kidney tumor growth in vivo. Mechanistically, this study identifies CBF-B as a potential therapeutic target for the treatment of kidney cancer and identifies novel biological insights into the regulation of the STING pathway in kidney cancer. However, the manuscript requires further clarification and revision before consideration for publication. Please see the detailed comments below.

We thank the reviewer for their appreciation of our studies. Our responses to their helpful suggestions are outlined below.

Figure 1- CBF-B was identified as a synthetic lethal partner of VHL loss in kidney cancer cells. Why is CBF-B loss only lethal only in VHL null cells? The expression levels appear to be similar, but the transcriptional activity appears to be different in the VHL proficient deficient setting.

We apologise for the confusion here. Synthetic lethality does not require different levels of expression, but relies on the combined loss of two genes increasing cell death, as classically exemplified in the cancer field with BRCA mutations and PARP inhibition. In our case, as we acknowledge in the text, we do not have the full explanation for why VHL and CBF- β are synthetic lethal, but the main cellular response altered is through type I interferon signalling. As outlined to reviewers 2 and 3, we think that the relatively high levels of cytosolic dsDNA from mitochondrial leakage following *CBFB* loss in *VHL*-null cells may contribute to this phenomenon (**Extended Data Fig. 6d**). It is likely that many of the transcriptional differences which the reviewer highlights are a result of HIF signalling in *VHL*-null cells, although we find that both the ISG effect, and the lethality phenotype itself are HIF-independent.

Figure 2a- The Kaplan-Meier curve showing CBF-B mRNA expression correlates with poor survival in the KIRC cohort would be nicely complemented with data showing if CBF-B protein levels are either increased in VHL deficient kidney cancer compared to normal kidney tissue and/or if CBF-B protein levels correlate with patient outcomes.

We thank the reviewer for this helpful suggestion. We have analysed data from the CPTAC ccRCC Discovery study which demonstrates a similar relationship between CBF- β protein levels and patient survival to that already described for its mRNA expression (**Fig. 2c**). Moreover, we found that CBF- β is overexpressed in ccRCC tumours relative to adjacent normal kidney tissue (**Fig. 2a**).

Figure 2g- The kinetics of the subcutaneous tumor growth curves would be important to show here.

We opted to include the data for day 28 alone for simplicity, but have now included the full growth curve as suggested (**Fig. 2h**). Tumours in this model show an initial growth over the first two weeks before regressing slightly and then adopting a sustained growth trajectory (see PMID: 34155378 Fig. 4n for a previous example of this). We have explained this in the updated figure legend. We elected to halt the experiment at day 28 post-injection in order to obtain tissues for immunohistochemistry from the *CBFB* knockout tumours before they fully regressed, but unfortunately there was too little tissue remaining to undertake this. We were, however, satisfied that the difference in tumour area between the control and knockout tumours seen at both day 21 and day 28 indicated a significant impairment of tumour establishment. These results were strongly supported by subsequent data obtained from the orthotopic xenograft model (**Fig. 2i-m** and **Extended Data Fig. f**)

Figure 3- Does rescue of RUNX expression restore growth and survival in the CBF-B knockout cells? This would be important to determine if RUNX proteins are functionally involved in the CBF-B growth and survival phenotype.

We thank the reviewer for this helpful suggestion. 786O cells do not tolerate RUNX overexpression well, and expression causes a significant growth delay in routine culture. Therefore, it has not been possible to directly measure if they restore growth. Nevertheless, we have now confirmed that RUNX proteins do indeed play an important role in ISG expression, with RUNX1 overexpression reducing ISG expression in *CBFB* depleted cells (**Fig. 7c,d**). Therefore, RUNX proteins and CBF- β are functionally involved in the same pathway.

Figure 3-4- Is it thought that RUNX is also involved in the CBF-B regulation of the ISG? It would be helpful to link the RUNX story to the STING story with data if they are proposed to be linked, or are they separate mechanisms?

We are sorry that we were not sufficiently clear with our explanations. We do think that RUNX is also involved in the regulation of ISGs and have clarified this further with additional overexpression experiments (**Fig. 7**).

Figure 5h- A better comparison here would be to compare the sensitivity of the 786O to the 786O+VHL cells. Are the VHL deficient cells more sensitive to IFN- β mediated cell death?

While HK2 cells give a more faithful representation of normal kidney tissue, we agree that there is value in investigating the IFN- β sensitivity of VHL-reconstituted 786O cells, and have included this data in **Extended Data Fig. 5l**. VHL loss mildly increased sensitivity to IFN- β with regards to cell death, consistent with our findings that IFN signalling has a propensity to result in a fitness disadvantage in VHL-deficient tumours.

Figure 5 extended data- Does CBF-B loss in VHL deficient kidney cancer cells result in increased numbers of γ -H2AX foci formation and DNA damage? The number of foci rather than total signal is important to rule out that CBF-B loss does not induce DNA damage.

Thank you for this helpful point. We have now also examined γ -H2AX foci formation by immunofluorescence as suggested, but again have not discerned any increase in DNA damage following CBF- β loss in *VHL*-null cells (**Extended Data Fig. 6e,f**).

Discussion- Can you please comment on the therapeutic potential of targeting CBF-B for cancer therapy. Are there current inhibitors, what is the potential toxicity of systemic inhibition?

Thank you for this suggestion and we have now included relevant comments in the Discussion.

Response to Reviewers

Reviewer #2 (Remarks to the Author):

In this revision, the authors have incorporated new data in an effort to address several key points, and the manuscript reflects some improvement over the previous version. However, substantial concerns regarding the proposed model remain unaddressed, and issues related to data interpretation and internal inconsistencies persist. The following points outline these unresolved concerns:

Overview of the main claims in this revision

The revised manuscript presents a model summarized in Figure 8, which includes several major claims:

Claim 1: CRISPR screening reveals that depletion of CBF- β is synthetically lethal in the context of VHL loss in clear cell renal cell carcinoma (ccRCC).

Claim 2: Loss of VHL leads to increased levels (or dosage) of CBF- β (as shown by comparing the middle and right panels in Figure 8), which is proposed as a partial mechanism underlying the synthetic lethality.

Claim 3: CBF- β directly represses STING, which modulates the type I interferon pathway. CBF- β binds directly to the STING locus, and although RUNXs are involved, the interaction between CBF- β and RUNXs is not required for this repression.

Claim 4: The authors propose that the STING–TBK1–IRF3 axis directly regulates the expression of interferon-stimulated genes (ISGs). They argue that STAT1/STAT2 are not involved in ISG expression; instead, IRF3 directly induces ISGs.

This reviewer has no major concerns regarding Claim 1, which is supported by the screening results and validation data showing that CBF- β depletion preferentially kills VHL-null cells.

We believe the reviewer has misunderstood several aspects of our work, leading to a characterisation of claims that differ from our actual conclusions.

Major Concerns

1. Claim 2 — that loss of VHL increases the dosage of CBF- β (Fig. 8)— appears weak and is contradicted by the data. It seems the authors propose this mechanism to partially explain the observed synthetic lethality. However, if that is the case, the data presented in Extended Data Figures 1d, 2d, and 3c do not support the claim. In fact, CBF- β levels are slightly lower in VHL-null cells compared to VHL-reconstituted cells, which directly contradicts the proposed model.

We wish to clarify that we do not claim VHL loss increases CBF- β dosage. We represented VHL increased dosage in Fig. 8 as there is clear evidence presented in Fig. 2a that CBF- β is overexpressed in ccRCCs compared to normal kidney tissue, explaining why this mechanism may enable specific targeting of cancer cells. The reviewer appears to be confusing a central point of the paper, which is the synthetic relationship between CBF- β loss and VHL loss in ccRCC cells, with CBF- β dosage. We are happy to clarify this in Fig. 8.

2. Claim 3 — that CBF- β directly represses STING mRNA, and although RUNXs are involved, the interaction between CBF- β and RUNXs is not required for this repression — lacks logical coherence and is not adequately supported by the data. In Figure 3, the authors attempt to demonstrate that the interaction between CBF- β and RUNXs is not required for ISG induction (Figures 3e, 3f, and 3g), yet both CBF- β and RUNXs are independently involved. The key unresolved question is: how can this be reconciled?

The authors propose several possibilities. For instance, if CBF- β merely regulates RUNX translation, as suggested in their cited work (Malik, 2019), and RUNX proteins can repress

STING and ISGs independently of CBF- β , then the model shown in Figure 8 is inaccurate or unsupported — it depicts RUNXs recruiting CBF- β to the STING locus. Given that CBF- β is an indispensable binding partners of RUNX, this possibility is highly unlikely, if not impossible.

Moreover, in this revision, the authors claim that CBF- β directly binds to the gene body of STING. However, since CBF- β lacks a DNA-binding domain, it remains unclear how it is recruited to this site since the authors now try to claim that the interaction between CBF- β and RUNXs is dispensable for the regulation. Do RUNXs bind to the STING gene body in ccRCC cells? If so, why is their interaction with CBF- β not required?

As it stands, Claim 3 is not convincingly supported by the data presented.

We respectfully disagree with the reviewer's interpretation of our data and conclusions. We explain in the manuscript (for example in lines 313-314 and the preceding paragraph) that RUNX1 appears to be the dominant regulator of STING expression in this context, but that the presence of CBF- β enhances its effect at a chromatin level. This conclusion is well-supported by the data, as we previously explained to the reviewer in response to point 2 of their original review: '[our findings suggest] that RUNX1 (and to a lesser extent RUNX2) is indeed the primary effector of STING repression, but that CBF- β dimerisation is likely to be required for maximal efficiency. This is also consistent with the small difference between WT and N104A CBF- β in the figure highlighted by the reviewer (Fig. 4e).'

There are other inaccuracies in the reviewer's response to this claim. First, Fig. 3 makes no allusion to ISG induction, so we are unable to respond to the specific challenge regarding this figure. Second, as we explain in the text (lines 358-359), CBF- β binds RUNX, but can perform RUNX-independent roles. Therefore, the model depicted in Fig. 8 is well-supported by the data we present, and the established body of RUNX/CBF- β literature. We do not claim at any point that CBF- β binds DNA directly, but show that there is a dependency on RUNX at the chromatin level, and confirm that CBF- β overexpression can influence RUNX translation.

3. Claim 4 — that IRF3 directly induces ISGs and that STAT1/STAT2 are not involved — requires more robust experimental support. To demonstrate direct regulation by IRF3, the authors performed conventional ChIP assays (Extended Data Fig. 5g). However, the signal (% of input) was low and nearly indistinguishable from background noise. Among the three genes tested, only IFIT2 showed a marginal yet statistically significant enrichment. Importantly, IFIT1 — a central model gene used throughout the manuscript — showed no detectable difference, raising doubts about whether IRF3 directly regulates IFIT1 or other ISGs.

At present, the data only establish that IRF3 is required for ISG expression, without clarifying the underlying mechanism. To substantiate the claim of direct regulation, the authors should perform IRF3 ChIP-seq, which would offer a more comprehensive and convincing assessment of IRF3 binding across ISG loci.

In addition, it is worth noting that STING has been shown to activate noncanonical NF- κ B signaling, which promotes cell survival and metastasis (see Bakhoum et al., Nature: "Chromosomal instability drives metastasis through a cytosolic DNA response", 2018). This alternative pathway should be considered in the interpretation of STING's role in ISG regulation and tumor biology.

We believe the reviewer has overlooked established literature, as referenced in our manuscript (refs. 54-56), which provides considerable evidence that IRF3 can directly induce ISGs. We also show in the revised manuscript binding of IRF3 to ISREs of key ISGs. We believe additional ChIP-seq experiments are unnecessary given the substantial supporting evidence in the established literature.

Furthermore, the paper to which the reviewer alludes is not relevant in this context: the authors are explicit that the activation of non-canonical NF- κ B signalling by cGAS-STING in CIN-high cells **does not** result in IRF3 phosphorylation or ISG induction (Barkhoum et al., 2018: Extended Data Figs. 8 and 9).

We present robust evidence that the induction of ISGs is IRF3-driven, both by direct dissection of this pathway, and the exclusion of other plausible mechanisms (e.g. IFN- β expression and JAK/STAT signalling).

4. Regarding the new ChIP-seq data for CBF- β , the results appear highly unusual. Across seven chromosomes, only a few peaks were detected, which is atypical. A quick literature search indicates that ChIP-seq datasets for RUNX proteins and CBF- β typically yield thousands to tens of thousands of peaks. For a transcriptional regulator, such sparse peak detection often suggests a failure in achieving specific or efficient ChIP enrichment. Unfortunately, it is not currently possible to assess the quality of this dataset, as the ChIP-seq data (GSE270776) remain private, even when accessed with the provided token: upszseuqvbyddsp. In contrast, the CRISPR screening dataset (GSE270775; token: anwvmykobbsttcl) is accessible.

To better evaluate the quality of the CBF- β ChIP-seq data, could the authors clarify the total number of peaks detected using an enrichment threshold of 5 and FDR < 0.01? This information would be critical for assessing the reliability and interpretability of the dataset.

The reviewer originally asked for CBF- β ChIP-seq to align to results we showed with RUNX publicly available data and the CBF- β ChIP-PCR experiments. We provided this with the full knowledge that ChIP-seq of a protein that does not directly bind DNA will result in sparse peaks. This was entirely expected. The finding that the *STING* locus was the dominant site for CBF- β points towards this region as having high RUNX/CBF- β occupancy, as we already discuss in the text.

We believe the reviewer is mistaken regarding the availability of CBF- β ChIP-seq datasets. We have investigated thoroughly, and not found any ChIP-seq data for CBF- β alone (ChIP-seq data only exists for the CBF- β -MYH11 fusion protein, which is entirely different oncogenic fusion protein). Many RUNX ChIP-seq datasets exist, but as previously stated, ChIP-seq of CBF- β will differ as it does not directly bind DNA.

We are unclear why the ChIP-seq link did not work but we have verified that the new link below is working (we would of course have provided this earlier in the review process if we had been alerted to the issue at the time). We also include the PCA plot below showing clustering of CBF- β binding compared to the inputs, a table of the number of fragment peaks, and how we calculated the peaks with enrichment threshold > 5 and FDR < 0.01.

<https://www.ncbi.nlm.nih.gov/geo/query/acc.cgi?acc=GSE300828&token=kpoxqwiejbenjqn>

PCA plot for ChIP-seq. Empty Vector (EV) and CBF- β overexpression conditions, with inputs, shown.

Sample	Total fragments	Fragments in peaks
EV1	143951416	18738
EV2	105296866	6405
EV3	153906230	24475
OE1	134123166	9169
OE3	143404056	30043

Total number of aligned fragments and number of fragments in peaks called by MACS. The Empty Vector (EV) condition has one consensus peak with fold change > 5 and FDR < 0.01, while the Overexpression (OE) condition has 5. In both these conditions the highest fold change was observed at the STING locus (15.6 and 39.9, respectively). We calculated the total number of aligned fragments and the number of fragments in peaks per replicate (Table 1). The numbers of reads in peaks is low, but CBF- β is not known to bind DNA directly, so we were most likely only able to detect loci with intense CBF- β chromatin binding (i.e. STING) in our ChIP-seq experiment.

5. Regarding Major Concern 1 (as raised in the rebuttal), this issue remains unaddressed, possibly due to a miscommunication. While it is acceptable that type I IFN does not fully explain the observed synthetic lethality—as the authors have acknowledged—this was not the central point of the previous review.

The key concern is that the authors claim CBF- β knockout is synthetically lethal in VHL-deficient cells and CBF- β directly regulates STING. Despite the efforts to establish both types of regulation, the authors have not provided any data connecting the synthetic lethality to type I IFN regulation. This gap is particularly significant given that both synthetic lethality and type I IFN signaling are emphasized in the title and represent two major themes of the study.

As a transcriptional regulator, CBF- β influences a wide array of cellular pathways. Without genetic evidence directly linking type I IFN signaling to the synthetic lethality phenotype, the observed regulation may be incidental. To address this, the authors could perform cell death and in vivo tumor assays—such as those shown in Fig. 2—using STING and IRF3 wild-type and knockout VHL-null ccRCC cells, which should be straightforward to implement.

In its current form, the manuscript presents two distinct regulatory themes—synthetic

lethality and type I IFN signaling regulated by CBF- β —without establishing a mechanistic connection between them.

The reviewer recognises that ‘it is acceptable that type I IFN does not fully explain the observed synthetic lethality – as the authors have acknowledged’. Given the reviewer’s acknowledgement of this point, we believe additional experiments to establish this connection are beyond the scope of this study. As previously explained, investigated in **Extended Data Fig. 5 i-k**, and stated in the text, the two mechanisms are related but the changes in type I IFN secretion with *CBFB* loss are not sufficient. We made this very clear at the outset of our first submission.

6. Furthermore, in rebuttal point 1, the authors propose: “It is therefore plausible that VHL-null cells may have a basal degree of mitochondrial instability, rendering them susceptible to mitochondrial insults and activation of the cGAS-STING pathway (in the context of *CBFB* knockout).” However, the more pertinent and unresolved question is how *CBFB* knockout activates cGAS, a cytosolic sensor of double-stranded DNA.

The authors state that *CBFB* knockout does not lead to mitochondrial dysfunction or genomic DNA damage (new Extended Data Fig. 6a–h). In the absence of evidence for DNA leakage or damage, the mechanism underlying cGAS activation remains unclear. This gap is critical, as it directly affects the interpretation of how *CBFB* loss engages the cGAS-STING pathway in VHL-null cells and undermines the proposed model.

We disagree with this assessment. We show the increase *STING* expression occurs regardless of cGAS (see point 7 below). Further studies on cGAS are not critical to our mechanism and clearly beyond the scope of what is already a substantial manuscript.

7. Regarding rebuttal point 3, the authors present data in Fig. 6e (right panel) showing that cGAS sgRNA reduces *STING* mRNA induction. However, this appears to be a red herring. The appropriate approach to assess whether cGAS regulates *STING* activation is to perform a cGAS knockout and examine phosphorylated *STING* or *TBK1*, as cGAS primarily regulates *STING* post-transcriptionally, consistent with the signal transduction model illustrated in Fig. 6a and Fig. 8.

As currently presented, the manuscript begins with a mechanistic framework in Fig. 6a, yet diverges from it by investigating transcriptional regulation of *STING* mRNA by cGAS. If the authors intend to demonstrate that cGAS regulates *STING* transcriptionally, this contradicts the premise of Fig. 6a and reflects a conceptual inconsistency. Conversely, if they conclude that cGAS does not regulate *STING* mRNA, this conflicts with the data shown in Fig. 6e (right panel), which indicates a statistically significant reduction.

In either case, the data concerning cGAS are problematic, and the mechanism of *STING* activation remains insufficiently addressed. Moreover, the authors’ interpretation does not resolve the discrepancy between their conclusions and the presented data. In the text (lines 292–293), they state: “cGAS knockout did not affect *CBFB*- β knockout-induced *STING* expression (Fig. 6e and Extended Data Fig. 6i).” Yet Fig. 6e (right panel) clearly shows a statistically significant difference ($p=0.012$), which contradicts this claim.

We believe there has been a misunderstanding regarding the purpose of **Fig. 6e** and the relationship to other experiments. The purpose of this panel is not to demonstrate that cGAS regulates *STING* transcriptionally, but the opposite. This experiment shows that the **increase in *STING* transcription upon *CBFB*- β knockout is in fact almost completely preserved upon cGAS knockout**, therefore showing that the effect of *CBFB*- β on *STING* is direct and not mediated by IFN signalling. We recognise that there is a minor reduction in *STING* expression in the right-hand column, as would be expected given that *STING* is itself an ISG, and would be willing to qualify our statement in the text to clarify this.

8. In this revision, the authors attempt to conclude that IFN- β (type I IFN) is not involved and propose a model for the direct regulation of ISGs by IRF3 (Fig. 5). However, the data presented—such as those concerning STAT1/STAT2—are suggestive but not definitive. Direct evidence would require demonstrating that IFNB gRNA has no effect, which has not been shown. JAK kinases have been shown to activate other pathways, such as PI3K-AKT and RAS–MAPK Pathway.

As explained above in response to point 3, we have presented considerable evidence that IRF3 directly regulates ISGs. It is important to note that IFN- β is not induced by *CBFB* knockout unless HT-DNA is also applied to the cells (Fig. 6d, Extended Data Fig. 7). IFN- β therefore cannot be secreted in response to CBF- β loss alone, and so any further exploration of JAK signalling pathways would be futile.

Minor Points:

1. In Figure 3, the objective and the title is misaligned. The primary goal of Figure 3, as stated in the text (line 163), is to demonstrate that both CBF- β and RUNX proteins contribute to the synthetic lethal interaction with VHL. However, the figure title seems to focus on how RUNXs are regulated by CBF- β at the translational level. This emphasis appears to diverge from the main theme of the manuscript. As the authors themselves noted, the post-translational regulation of RUNXs by CBF- β has already been addressed in a previous publication (Malik et al., 2020, Nature Communications). It is recommended that the author stick to their main topic.

Our data on the regulation of RUNX at the translational level provide important validation of the previous findings of Malik et al. (to our knowledge this finding has not yet been reproduced by any other groups), and help to explain our results regarding RUNX and CBF- β . We can revise the title accordingly.

2. Clarification is needed for “Immune Cell-Specific IRF7”: The phrase “immune cell-specific IRF7” is ambiguous. Are the authors implying that IRF7 expression is restricted to immune cells? This interpretation appears inconsistent with their own data in Extended Data Figure 5e, which shows IRF7 expression in cancer cells. Further clarification is warranted to resolve this confusion.

We added the phrase ‘immune cell-specific IRF7 in response to Reviewer #3 (minor point 1), and note that Reviewer #3 is now satisfied with this amendment. We do not agree that our interpretation is inconsistent. As explained in ref. 57, IRF7 is only expressed at significant levels in immune cells; in the immunoblot (Extended Data Fig. 5e), there is only an extremely low level of expression in ccRCC cells, hence the very faint bands. Regardless of the terminology used, the key purpose of the experiment was to show that our mechanism was driven by IRF3 (Fig. 5c).

Reviewer #3 (Remarks to the Author):

Since the author has provided clarifications and additional experiments in response to the reviewers' concerns, the manuscript is now deemed acceptable for publication upon re-evaluate.

We thank this reviewer for their appreciation of our work and the recommendation for publication.

Reviewer #4 (Remarks to the Author):

The authors have addressed the majority of my concerns. However, Figure 2h should be shown as the tumor volume $\text{tumor volume (mm}^3) = (\text{length} \times \text{width}^2)/2$, not tumor area.

We thank this reviewer for the acknowledgement that we addressed their concerns, and can correct this this minor point.

Reviewer #5 (Remarks to the Author):

Bertlin et al describe a series of genetic screens ostensibly to discover genes that are essential in the context of VHL loss, the signal genomic lesion in kidney cancer (clear cell renal cell carcinoma, ccRCC). The authors perform genome-scale CRISPR knockout screens in two ccRCC cell lines, 786O and RCC4, which carry the signature homozygous VHL loss, and again in those same cells with exogenously expressed VHL.

It is not clear how well-controlled the CRISPR screens are. The authors use the TKOv3 one-component library. Some quality control figures are present in Supp Fig 1, but the relative paucity of hits (genes with Bayes Factor > 0 or some higher threshold) shown in Fig 1C suggests that quality was not optimal. It would be useful to have a precision-recall curve, using common reference sets of essential and nonessential genes (e.g. for essentials, Hart et al, G3, 2017 – the Hart et al, Cell 2015 set is not optimal – and for nonessentials, Hart et al MSB 2014). Other data, such as cell-line specific nonexpressed genes, may be substituted for nonessentials without loss of information. Comparison of the PR curves of these screens to those publicly available ccRCC genome-scale cell line screens in DepMap would provide further evidence of screen quality. This information is important, as it gives some insight into the false negative rate of a cell line screen, which is critical in evaluating whether a hit found in one screen but not another reflects underlying biology or just a missed call in the other screen.

We thank this reviewer for their assessment of our work. We did not include all the control figures, but please see below the precision-recall (PR) curves using the reference sets for non-essential and essential genes as suggested. These show the PR values improving over the duration of the screens relative to the sequenced plasmid, as described in Hart and Moffat, BMC Bioinformatics, 2016 (PMID: 27083490).

I am satisfied with this analysis for the PR curves.

I am wondering what the authors are plotting in Fig. 1C. Both axis say “Bayes Factor” and the cell line screen. It appears they are actually plotting $\Delta\text{BF}(\text{VHL}^{-/-}) - (\text{VHL}^{+/+})$ to find VHL SL interaction that are common to both cell lines. This will need to be clarified. If they are indeed plotting ΔBF then I agree, the SL interactions they identified are relatively weak since the ΔBF between VHL^{-/-} and VHL^{+/+} are relatively low. Since the remainder of the manuscript is validating CFBF then I think the screen holds for the purpose of this manuscript.

We thank this additional reviewer for commenting on the prior reviewer’s points, and agreeing that the screen identifies CFBF as a synthetic lethal interaction with VHL loss. We provide comments where the reviewer requested further clarification.

We have updated the Venn diagrams displayed in Extended Data Fig. 1c, to include the core essential gene set described in Hart et al., G3, 2017 (PMID: 28655737). Again, these show robust discrimination of essential genes.

I am satisfied with the Venn diagrams

We accept that there will be false negatives from our approach, but our aim was to identify potential synthetic lethal interactors of VHL which may be pertinent to ccRCC therapy, rather than to provide a comprehensive atlas of VHL-associated synthetic lethality.

I agree with the authors on this point, they are not looking to catalog all the SL interactions between with VHL-LOF, they are simply trying find some that are present in both cell lines which they have accomplished.

To identify genes providing proliferative advantage (e.g. KEAP1 in this case), a better approach is described in Lenoir et al, Nat Comms 2021. This is far better than Bagel scores, although the outlier nature of KEAP1 knockout is a strong signal here.

We have analysed our data with both DrugZ and MAGeCK to obtain enrichment scores for genes that improve survival. Please see the results below (also presented in Supplementary Table 1), which show that KEAP1 sgRNAs are by far the most enriched, regardless of the analysis workflow employed.

I am satisfied with the DrugZ and MAGeCK analysis.

The cherrypicking of target genes (Fig 1d) is useful for narrowing the reader's attention to the genes of interest but the display is not convincing. It is standard to calculate and display fold change of normalized end point cells relative to some early point (plasmid or T0), rather than read counts, alongside some negative control (again, nonessentials are useful here) whose distribution is centered around zero.

We elected to display the trajectory of normalised sgRNA read counts as this includes both the desired fold change information, as well as a central timepoint to provide additional temporal information about the depletion/enrichment of sgRNAs over the course of the screens. We have added the sgRNAs targeting EGFP as negative controls to Fig. 1d. As expected, these show minimal change, with only mild enrichment in RCC4 cells as a result of the dropout of essential genes from the pool.

I agree with reviewer 1 that displaying the normalized sgRNA read count trajectory is not typical. However, Fig. 1d is essentially another way to display the screen data and is not critical for the remaining scope of the manuscript.

For comparing differential essentials between cell lines, the approach is flawed. The difference in proliferation rates between the VHL+ and VHL- cells will confound a competition assay for single gene effects. A better approach would be to use an algorithm such as DrugZ (Colic et al, Genome Med 2019; see, e.g., Karasu et al Nat Biotech 2024 for a recent example of its application) to compare endpoint readcounts of screens in VHL+ and VHL- cells.

We had already used MAGeCK but chose to display the finding with BAGEL2. We have now analysed the data with DrugZ and neither of these additional methods alter our findings: as displayed below, CFBF remains a top hit within all of these analyses. Full results are given in Supplementary Table 1.

I am satisfied with the DrugZ analysis

The competition assays (Fig 1. f,g) are normalised to account for differential proliferation rates, and mCherry/GFP-expressing clonal cell lines were selected to minimise proliferation

differences. Furthermore, the fact that guides targeting CFBF and KEAP1 produce opposite effects in these assays further reinforces our conclusions in this initial validation experiment.

I am also not convinced by the pooled competition assay result. Reviewer 1 is correct, the differential growth rates of VHL^{-/-} vs. VHL^{+/+} will confound the assay and is very hard to control for. Typically, these types of competition assays are done with one cell line comparing control sgRNA to another sgRNA. Colony formation assays are a better way to validate the SL interaction when analyzing two isogenic cell lines with different proliferation rates and should be considered.

*To clarify, original reviewer 1 was concerned about the use of identifying essential interactions in the **screen** using two cell lines, hence the suggested to use DrugZ, which we show still identifies CFBF. They had no criticism of the competition assay in **Fig 1. f,g**. In fact, the competition assay is well established for looking at synthetic lethal interactions, including those for VHL (e.g. PMC6913182, Fig. 3, from the Kaelin Jnr group). It is not difficult to control for growth rates between the VHL ^{-/-} or VHL ^{+/+} cells. Additionally, the reviewer has overlooked **Fig 2d, which is already a colony formation assay and shows the synthetic lethal interaction between CFBF and VHL loss.***

My motivation for stressing controls so firmly is that the main gene chosen for follow up, CFBF, is a very weak hit in Fig 1d. More importantly, it is not a hit in DepMap. In fact CFBF knockout shows strong phenotype in leukemia, lymphoma, and myeloma cells (Chronos score < -0.5), but is never essential in any of the >25 kidney cancer cell lines screened by DepMap. The in vivo data is compelling, but is totally inconsistent with the in vitro data that were ostensibly used to nominate CFBF for validation.

We appreciate that this reviewer found the in vivo data compelling, but we respectfully disagree with their assertion regarding the screens. We set out to take an unbiased strategy to look for experimentally find synthetic lethal interactors, which is exactly why we used the dropout CRISPR approach. As would be important for any screening approach, we go on to validate our findings. The screening data is entirely consistent with the subsequent validation. Several genes and pathways known to be important in targeting renal cell cancer are not hits in DepMap (e.g. TBK1, ROCK1, FTO, SCARB1), and DepMap is not a substitute for undertaking a screen based on VHL deficiency or proficiency. As discussed previously (PMID: 34244212), differences between studies may arise due to the use of different cell culture media. Other possible reasons include the employment of different ccRCC cell lines and CRISPR/RNAi libraries in screening experiments.

While CFBF is a weak hit in the screen, I agree with the authors that lack of signal in DepMap does not invalidate their results. One possible explanation is due to the artificial nature of adding VHL back to cancer cell lines. While targeting CFBF in ccRCC may not be a feasible approach in the clinic, the authors have still found some potentially interesting biology to explore based on their screen results.

The authors should either perform a more realistic analysis of their screen data, or find another way to get to the CFBF hypothesis in ccRCC.

We disagree with this statement, and have performed a thorough analysis of the screen data. Experimental validation is always important following a screen: we have extensively demonstrated the lethal effect of CBF- β loss in ccRCC in vitro and in vivo, which is further supported by the overexpression of CFBF in ccRCC tumours.

I am satisfied with the reviewer response.

We thank this reviewer for the additional review of our work and that they are satisfied with the screening methodology and findings.

Response to Reviewer's Comments

Reviewer #6 (Remarks to the Author):

I find the manuscript to be technically well-executed, detailed, and to have identified a novel dependency in RCC that could be therapeutically relevant.

We thank the reviewer for their helpful feedback and appreciation of the work. We have addressed the remaining issues raised.

There are a couple issues that need to be addressed prior to publication:

Figure 8 does not adequately represent the findings of the manuscript as only the transcriptional function of CBF β is included. The figure should be modified to show both the transcriptional and translational roles (upregulation of RUNX1 in particular) of CBF β .

We now include the translational roles of CBF β on RUNX in Figure 8 and how this can feedback onto the CBF β /RUNX complex.

Fig. 8. Model of regulation of STING expression by CBF- β /RUNX

CBF β ChIPSeq has been performed and many peaks were identified (see for example <https://doi.org/10.1371/journal.pgen.1005884>) as Reviewer 2 indicated, so the authors' response to the critique about the lack of peaks in their CBF β ChIPSeq data is not compelling. Either the ChIPSeq should be repeated to achieve higher quality data or ChIP-qPCR should be done for the STING locus to confirm the presence of CBF β .

We thank the reviewer for highlighting the prior ChIP-seq data. To address their point we have analysed the ChIP-seq data from the highlighted study (done in SAOS2 osteosarcoma cells) and included ChIP-PCR of the STING locus (**Figure 7a-d**). Our ChIP-seq data showed binding at the distal end of the STING gene, whereas CBF β associated with the promoter region in SAOS2 cells. However, both show CBF β occupancy at the STING locus. We next used primers around the promoter and distal gene regions to further analyse CBF β occupancy in 786O cells. These 4 different primer sets confirm the presence of CBF β at the STING locus, with reduced binding following CBF β sgRNA-depletion (**Figure 7a-d**).

Fig. 7. RUNX/CBF- β proteins directly inhibit *STING* transcription